EMBO
*reports*

# EVL regulates VEGF receptor-2 internalization and signaling in developmental angiogenesis

Joana Zink[1,2,†], Maike Frye[3,*,†] [iD], Timo Frömel[1,2], Claudia Carlantoni[3], David John[2,4], Danny Schreier[3], Andreas Weigert[5], Hebatullah Laban[6], Gabriela Salinas[7], Heike Stingl[1,2], Lea Günther[1,2], Rüdiger Popp[1,2], Jiong Hu[1,2], Benoit Vanhollebeke[8] [iD], Hannes Schmidt[9], Amparo Acker-Palmer[10] [iD], Thomas Renné[3], Ingrid Fleming[1,2] [iD] & Peter M Benz[1,2,**,‡] [iD]

## Abstract

Endothelial tip cells are essential for VEGF-induced angiogenesis, but underlying mechanisms are elusive. The Ena/VASP protein family, consisting of EVL, VASP, and Mena, plays a pivotal role in axon guidance. Given that axonal growth cones and endothelial tip cells share many common features, from the morphological to the molecular level, we investigated the role of Ena/VASP proteins in angiogenesis. EVL and VASP, but not Mena, are expressed in endothelial cells of the postnatal mouse retina. Global deletion of EVL (but not VASP) compromises the radial sprouting of the vascular plexus in mice. Similarly, endothelial-specific EVL deletion compromises the radial sprouting of the vascular plexus and reduces the endothelial tip cell density and filopodia formation. Gene sets involved in blood vessel development and angiogenesis are down-regulated in EVL-deficient P5-retinal endothelial cells. Consistently, EVL deletion impairs VEGF-induced endothelial cell proliferation and sprouting, and reduces the internalization and phosphorylation of VEGF receptor 2 and its downstream signaling via the MAPK/ERK pathway. Together, we show that endothelial EVL regulates sprouting angiogenesis via VEGF receptor-2 internalization and signaling.

**Keywords** Ena/VASP proteins; endothelial cells; sprouting angiogenesis; tip cell filopodia formation; VEGF receptor 2 internalization and signaling
**Subject Categories** Cell Adhesion, Polarity & Cytoskeleton; Signal Transduction; Vascular Biology & Angiogenesis

## Introduction

The development of the vertebrate vasculature requires the concerted action of growth factors and guidance cues that target endothelial cells. In the angiogenic vasculature, the highly motile and invasive endothelial tip cells form actin-rich lamellipodia and filopodia, which probe the environment for guidance cues, such as vascular endothelial growth factor (VEGF), and thereby determine the direction of growth (Gerhardt *et al*, 2003; Carmeliet *et al*, 2009). VEGF binding triggers homo- and heterodimerization of endothelial VEGF receptors (primarily VEGFR2 and VEGFR3), which induces receptor tyrosine phosphorylation and the activation of downstream signaling pathways that control proliferation, migration, and sprouting (Carmeliet & Jain, 2011; Simons *et al*, 2016). Extracellular regulated kinase (ERK)1/2 is a prominent downstream target of VEGFR2 signaling and the phosphorylation of VEGFR2 on Y1173 (murine sequence; Y1175 human sequence) is crucial for ERK1/2 activation. Indeed, a non-phosphorylatable Y1173F VEGFR2 mutant has the same effect as VEGFR2 gene ablation, namely early embryonic lethality due to severe vascular defects (Takahashi *et al*, 2001; Sakurai *et al*, 2005). Given its crucial role in vascular development, the biological activity of VEGFR2 is tightly regulated, i.e., at the level of receptor expression, as well as by the presence of co-receptors and

1   Centre for Molecular Medicine, Institute for Vascular Signalling, Goethe University, Frankfurt am Main, Germany
2   German Centre of Cardiovascular Research (DZHK), Partner site Rhein-Main, Frankfurt am Main, Germany
3   Institute of Clinical Chemistry and Laboratory Medicine, University Medical Center Hamburg-Eppendorf, Hamburg, Germany
4   Insitute for Cardiovascular Regeneration, Goethe University, Frankfurt am Main, Germany
5   Institute of Biochemistry I-Pathobiochemistry, Faculty of Medicine, Goethe-University, Frankfurt am Main, Germany
6   Department of Cardiovascular Physiology, Institute of Physiology and Pathophysiology, Heidelberg University, Heidelberg, Germany
7   NGS-Integrative Genomics Core Unit (NIG), Institute of Human Genetics, University Medical Center Göttingen (UMG), Göttingen, Germany
8   Laboratory of Neurovascular Signaling, ULB Neuroscience Institute Department of Molecular Biology, University of Brussels, Walloon Excellence in Life Sciences and Biotechnology (WELBIO), Brussels, Belgium
9   Interfaculty Institute of Biochemistry, University of Tübingen, Tübingen, Germany
10  Institute of Cell Biology and Neuroscience and Buchmann Institute for Molecular Life Sciences, Goethe University, Frankfurt am Main, Germany
    *Corresponding author. Tel: +49 40 471057004; E-mail: m.frye@uke.de
    **Corresponding author. Tel: +49 69 6301 6052; E-mail: benz@vrc.uni-frankfurt.de
    †These authors contributed equally to this work
    ‡Present address: Department of CardioMetabolic Diseases Research, Boehringer Ingelheim Pharma GmbH & Co. KG, Biberach, Germany

auxiliary proteins, and the activity of signal terminating tyrosine phosphatases, e.g., CD148, PTP1b, and VE-PTP (Lampugnani *et al*, 2006; Lanahan *et al*, 2010; Hayashi *et al*, 2013; Simons *et al*, 2016). Although the surface expression of VEGFR2 is prerequisite for ligand binding, the endocytosis of the receptor is essential to activate many, if not all of the downstream signaling pathways, including ERK1/2 (Simons *et al*, 2016). A prominent example for this mechanism is ephrin-B2, which controls VEGF receptor internalization and is necessary for VEGF-induced tip cell filopodia extension and vascular sprouting (Sawamiphak *et al*, 2010; Wang *et al*, 2010). However, downstream mechanisms and proteins involved in VEGF receptor endocytosis and signaling have not been well defined.

Like axonal growth cones, the principal role of endothelial tip cells is to navigate, a process that requires the correct probing of microenvironmental cues and their translation into directed cell migration. Therefore, it is not surprising that molecular guidance cues are largely conserved between axonal growth cones and endothelial tip cells and both form filopodia to explore their local environment (Soker *et al*, 1998; Wang *et al*, 1998; Serini *et al*, 2003; Wang *et al*, 2003; Banu *et al*, 2006; Adams & Eichmann, 2010; Dent *et al*, 2011; Fischer *et al*, 2018). Despite their pivotal role in tip cell navigation, little is known about the processes regulating filopodia assembly in endothelial cells (Carmeliet *et al*, 2009; Fischer *et al*, 2018). Studies in axonal growth cones have highlighted the potential importance of the enabled/vasodilator-stimulated phosphoprotein (Ena/VASP) family for filopodia formation (Drees & Gertler, 2008). The Ena/VASP proteins are important mediators of cytoskeleton control, linking kinase signaling pathways to actin assembly and are localized at sites of high actin turnover, including cell–cell contacts, focal adhesions, the leading edge of lamellipodia, and the tips of filopodia, where they promote actin polymerization and regulate the geometry of F-actin networks (Krause *et al*, 2003; Sechi & Wehland, 2004; Benz *et al*, 2009). In axonal growth cones and fibroblasts, these proteins bundle actin fibers and antagonize the capping of elongating filaments, thereby promoting filopodia formation and cell motility (Lebrand *et al*, 2004; Schirenbeck *et al*, 2006; Applewhite *et al*, 2007; Dent *et al*, 2007; Drees & Gertler, 2008; Barzik *et al*, 2014; Winkelman *et al*, 2014). In mammals, the Ena/VASP family of proteins consists of mammalian enabled (Mena), VASP, and Ena-VASP-like protein (EVL). In endothelial cells, Ena/VASP proteins are required for intercellular adhesion and Ena/VASP gene disruption impairs vessel integrity and endothelial barrier function *in vivo* (Furman *et al*, 2007; Benz *et al*, 2008). However, a potential role of Ena/VASP proteins in tip cell filopodia formation and angiogenesis is not known. Given the structural and molecular similarities between axonal growth cones and endothelial tip cells, this study set out to investigate the role of Ena/VASP proteins in angiogenic sprouting *in vitro* and *in vivo* in the postnatal murine retina.

## Results

### Differential expression of Ena/VASP protein family members in postnatal retinal endothelial cells

To address potential roles of Ena/VASP proteins in angiogenic sprouting, expression of VASP, EVL and Mena was assessed in endothelial cells isolated from the postnatal murine retina. CD31 and CD34

double-positive endothelial cells were isolated by FACS from retinas from wild-type mice on postnatal day 5 (P5), and subjected to RNA-Seq (Fig EV1). Analysis of the transcriptomes confirmed high expression levels of multiple endothelial biomarkers, including von Willebrand factor (vWF), tyrosine-protein kinase receptor Tie2 (TEK), VE cadherin (CDH5), endoglin (ENG), CD146 (MCAM) and VEGFR2 (KDR). Endothelial selective sorting was confirmed by expression analysis of marker genes for astrocytes, immune cells, Müller glial cells, neurons and retina pigment epithelial cells, which were either very low or undetectable in the isolated cells (Fig 1A). Notably, while P5 retinal endothelial cells expressed VASP and EVL, Mena RNA was hardly detectable (Fig 1B). Consistent with expression data, confocal microscopy confirmed VASP and EVL protein expression in endothelial cells of the postnatal mouse retina (Fig 1C and D). No Mena was detectable in these cells, arguing against a significant role of Mena in retinal endothelial cells. However, our antibodies readily detected Mena in other retinal cells, including neurons (Fig 1E).

### EVL expression in endothelial cells

Based on the transcriptomic and immunofluorescent analyses, we focused on the *in vivo* roles of VASP and EVL for vascular sprouting. "Knockout first" mutant EVL mice were obtained from the European Mouse Mutant Archive (Skarnes *et al*, 2011) and used to generate global EVL-deficient (EVL$^{-/-}$) mice (Fig EV2A). Alternative splicing generates two EVL isoforms, a short (393 amino acids) and a 21 amino acids longer variant (EVL-I, (Lambrechts *et al*, 2000)). Western blotting using a novel polyclonal antibody raised against the murine EVL protein (Fig EV2B and C) detected both the EVL and EVL-I isoforms in lung and spleen lysates from adult wild-type mice and in the P5 brain and retina. However, only the short EVL isoform was detected in adult wild-type brain and retina and neither of the EVL isoforms were detectable in tissues derived from EVL$^{-/-}$ mice (Fig EV2D). Consistent with the Western blot analysis, flow cytometry of CD31$^+$CD34$^+$ cells from P5 mouse retinas revealed strong EVL expression in wild-type but not EVL$^{-/-}$ endothelial cells (Fig 2A). VASP is highly expressed in endothelial cells and required for endothelial barrier function *in vivo* (Furman *et al*, 2007; Benz *et al*, 2008; Benz *et al*, 2009; Kraft *et al*, 2010), but little is known about EVL function in endothelial cells. We analyzed EVL expression in endothelial cells from different human and mouse vascular beds. Both EVL isoforms were detected in the human endothelium-derived cell line EA.hy926, but only the short EVL protein isoform was detected in primary human dermal lymphatic endothelial cells (HDLEC) (Fig EV2C). Murine brain (MBEC) and lung (MLEC) endothelial cells isolated from wild-type mice also expressed both EVL protein isoforms but as expected no EVL protein signal was detectable in corresponding endothelial cells from EVL$^{-/-}$ mice (Fig 2B and C). Similar to the subcellular distribution of VASP in endothelial cells (Benz *et al*, 2009), EVL was concentrated at actin stress fibers, focal adhesions, filopodia, and the leading edge of lamellipodia in migrating MLEC (Figs 2D and EV3A).

### Impact of individual and combined VASP/EVL deletion on postnatal retinal angiogenesis

To study the roles of VASP and EVL for sprouting angiogenesis *in vivo*, we compared the postnatal development of the retinal

   

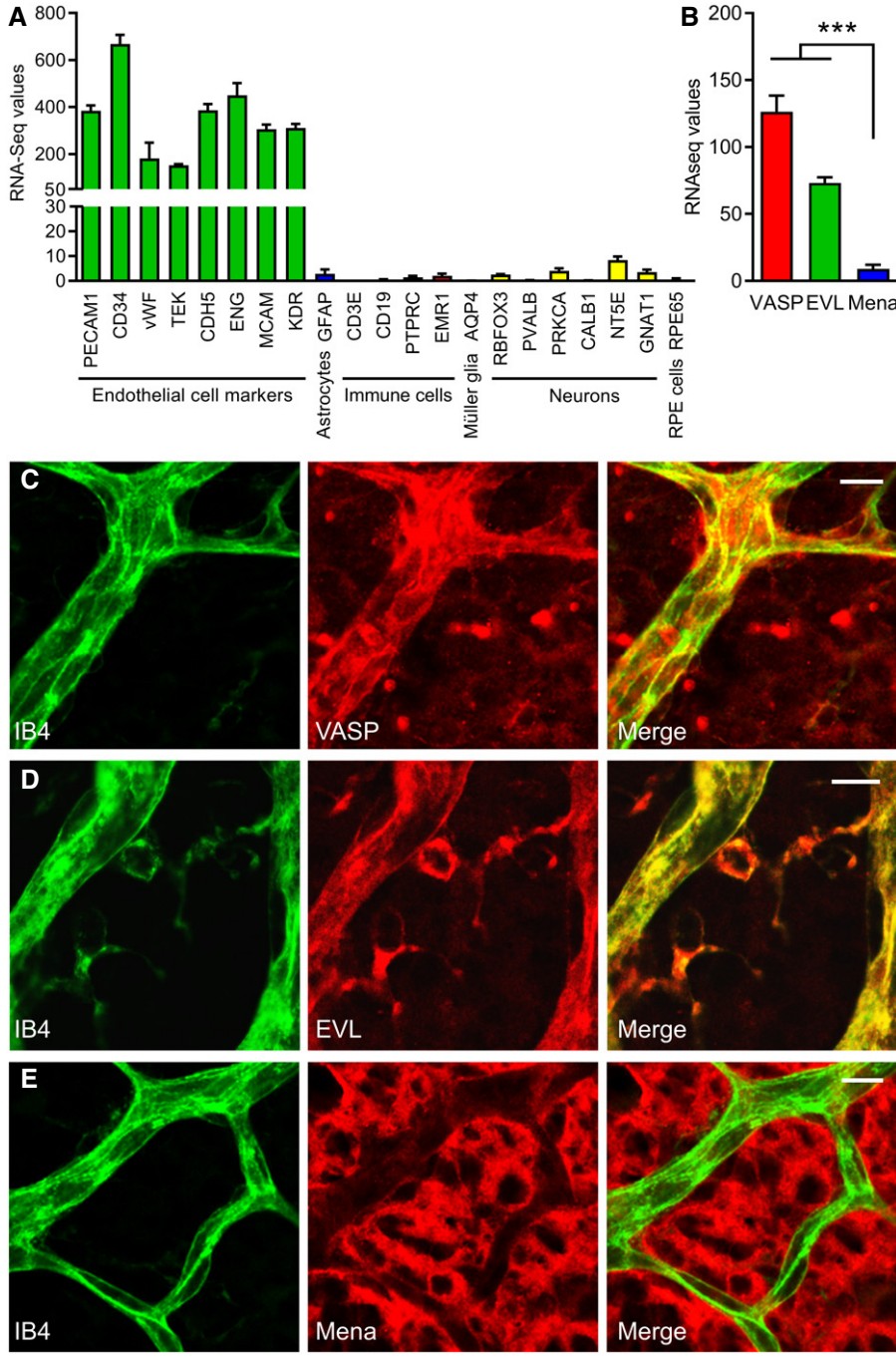

**Figure 1. Expression of Ena/VASP proteins in postnatal retinal endothelial cells.**

A, B RNA sequencing of CD31 and CD34 double-positive retinal endothelial cells from P5 wild-type mice. (A) RNA levels (FPKM; fragments per kilobase million) of marker genes of endothelial cells (green; CD31 (gene name PECAM1), CD34, von Willebrand Factor (vWF), tyrosine-protein kinase receptor Tie2 (TEK), VE-cadherin (CDH5), endoglin (ENG), CD146 (MCAM) and VEGFR2 (KDR)), astrocytes (blue, GFAP), immune cells (red; T cells (CD3E), B-cells (CD19), all leukocytes (CD45, PTPRC), and monocytes/macrophages (F4/80, EMR1)), Müller glial cells (orange; aquaporin 4 (AQP4)), neurons (yellow; retinal ganglion cells (RNA binding fox-1 homolog 3, RBFOX3), amacrine cells (parvalbumin, PVALB), bipolar cells (PKC-α, PRKCA), horizontal cell (calbindin, CALB1), photoreceptors (rods, CD73 (NT5E); cones, transducing (GNAT1)), and retinal pigment epithelial cells (magenta; retinal pigment epithelium-specific 65 kDa protein (RPE65)). (B) RNA levels of VASP (red), EVL (green), and Mena (blue) in the CD31 and CD34 double-positive P5 retinal endothelial cells. Error bars represent SEM; $n = 18$ animals (36 retinas) from six independent litters; four independent experiments. ***$P < 0.001$, one-way ANOVA with Bonferroni's multi comparison test.

C–E Staining of Ena/VASP proteins in blood vessels of P5 mouse retinas. P5 wild-type mouse retinas were fixed and stained with isolectin B4 (IB4, green) to visualize endothelial cells and VASP- (C), EVL- (D), or Mena-specific (E) antibodies (red). Yellow color in the merged images indicates the expression of VASP and EVL proteins in endothelial cells. No Mena protein expression was detected in P5 retinal endothelial cells (E). Representative images from three independent experiments are shown. Scale bars, 10 μm.

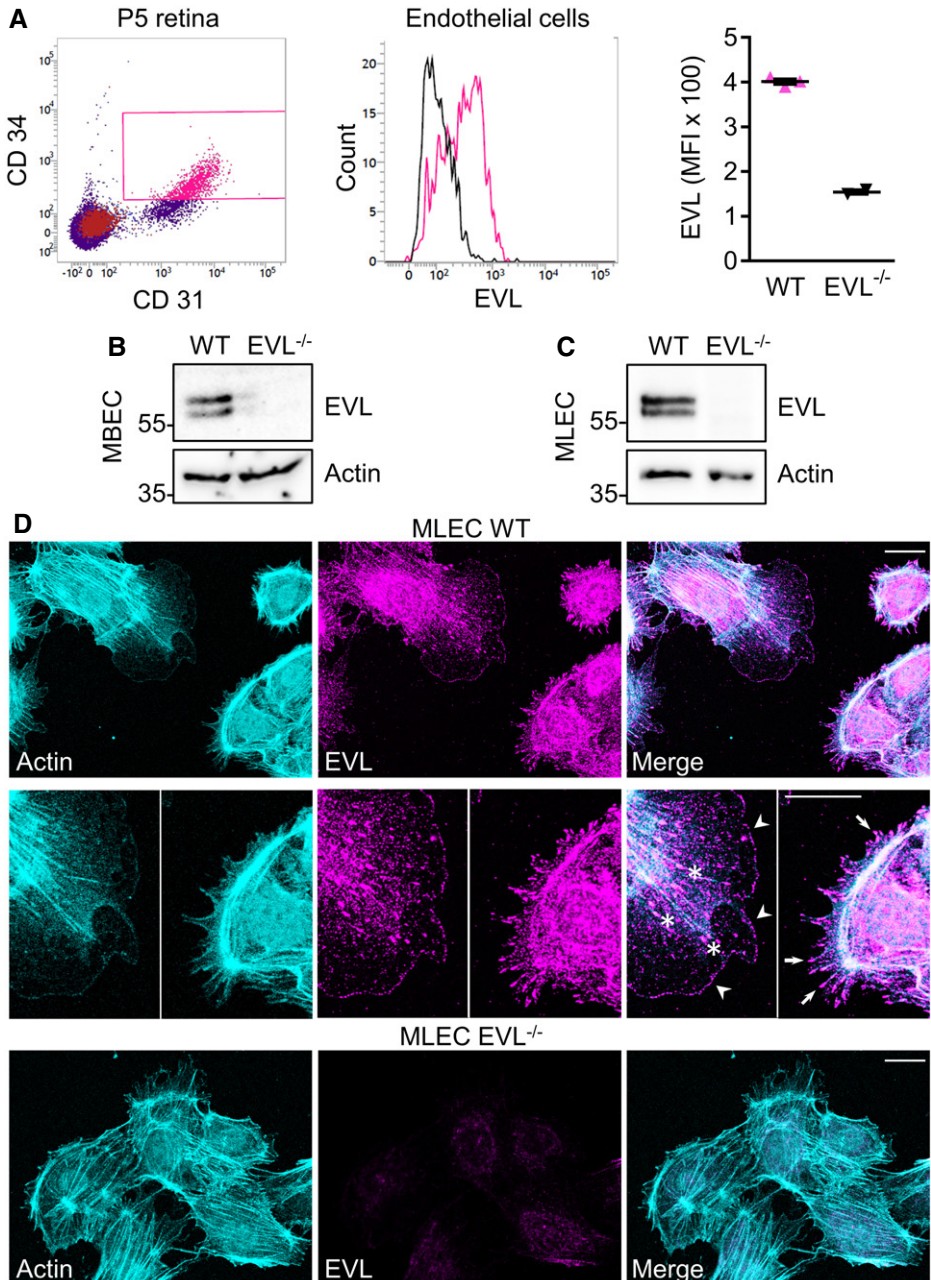

**Figure 2. EVL expression in mouse endothelial cells.**

A  Wild-type and EVL$^{-/-}$ P5 retinas were digested, labeled with CD31-, CD34-, and EVL-specific antibodies, and analyzed by flow cytometry. Endothelial cells were defined as CD31/CD34-double-positive cells (gate indicated in the left panel) and analyzed for EVL expression (middle panel). Mean fluorescence intensity (MFI) of EVL in wild-type (magenta) and EVL$^{-/-}$ (black) cells is shown in the right panel. WT: $n = 3$, EVL$^{-/-}$: $n = 2$; two retinas per animal. Error bars represent SEM.

B, C  EVL protein expression in sparse/migrating wild-type (WT) and EVL$^{-/-}$ mouse brain endothelial cells (MBEC) and mouse lung endothelial cells (MLEC). Actin was used as loading control. Western blots are representative of three independent experiments.

D  MLEC from wild-type (upper panel) and EVL$^{-/-}$ (lower panel) mice stimulated with 10 ng/ml VEGF were stained for actin (cyan) and EVL (magenta). Asterisks indicate focal adhesions, white arrows indicate filopodia, and white arrowheads indicate the leading edge of lamellipodia. Representative images from four independent experiments are shown. Scale bar, 20 μm.

vasculature in VASP$^{-/-}$ (Hauser *et al*, 1999) and EVL$^{-/-}$ mice. The radial extension of the vascular plexus from the optic nerve to the periphery at P3 and P5 was significantly delayed in EVL$^{-/-}$ mice. The magnitude of the delay inversely correlated with the vascularization of the retina; e.g., the delay was most pronounced in P3 retina, smaller in the P5 retina and vanished in the mostly normoxic P7 retina (Fig 3A and B), suggesting a role of EVL particularly in the early stages of hypoxia/VEGF-driven retinal

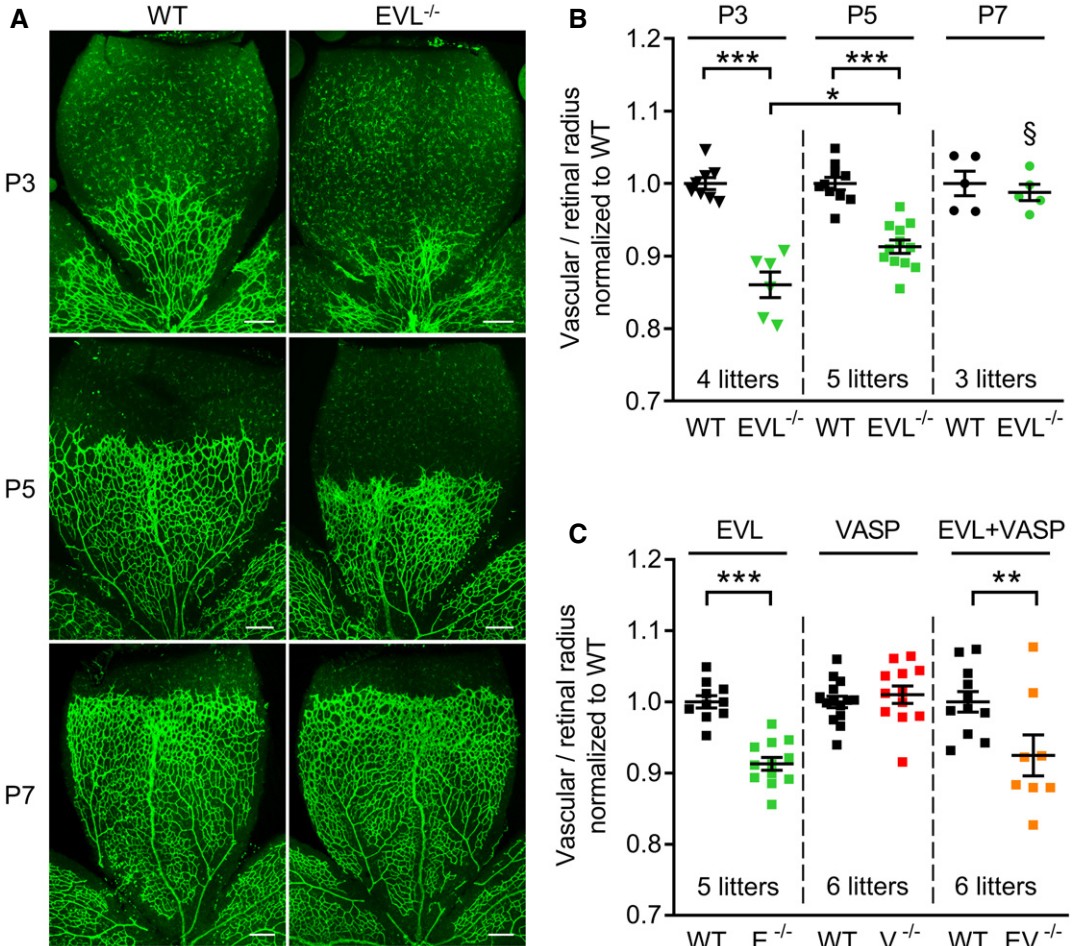

**Figure 3. Delayed postnatal retinal angiogenesis in EVL$^{-/-}$ mice.**

A   Isolectin B4 stained vasculature in whole mount retinas of wild-type (WT) and global EVL$^{-/-}$ mice on postnatal days 3, 5, and 7 (P3, P5, P7) assessed by confocal microscopy. Scale bars 200 μm.
B   Analysis of the radial vascular outgrowth relative to retinal radius and normalized to wild-type littermates.
C   Impact of individual and combined VASP/EVL deletion on sprouting angiogenesis in the postnatal mouse retina at P5 (EVL-deficient animals (E$^{-/-}$), green; VASP-deficient animals (V$^{-/-}$), red; VASP/EVL-double deficient animals (EV$^{-/-}$), orange).

Data information: Error bars represent SEM; *$P < 0.05$, **$P < 0.01$, ***$P < 0.001$, $^{§}P < 0.001$ vs. P3 and P5 EVL$^{-/-}$ (one-way ANOVA with Bonferroni's multi comparison test).

angiogenesis. In contrast, VASP deficiency had no significant impact on the rate of the vascular sprouting in the postnatal retina (Fig EV3), and mice with combined EVL/VASP deficiency were not statistically different from EVL$^{-/-}$ animals (Fig 3C).

**Endothelial-specific EVL deletion impairs vascular sprouting *in vivo***

Apart from endothelial cells, many other cell types contribute to the postnatal angiogenesis *in vivo*, including astrocytes, pericytes, and microglia. To specifically address the contribution of endothelial EVL to defective angiogenesis in the postnatal retina, we generated mice with endothelial cell-specific deletion of EVL (EVL$^{ΔEC}$) by crossing floxed EVL mice (EVL$^{fl/fl}$) with a tamoxifen-inducible, pdgfb-driven Cre deleter mouse (Fig EV2A). Similar to global EVL$^{-/-}$ mice, postnatal retinal angiogenesis at P5 was significantly delayed in EVL$^{ΔEC}$ pups compared with their wild-type (EVL$^{fl/fl}$)

littermates and no significant difference was observed comparing the radial sprouting in global and endothelial-specific EVL knockout mice (Fig 4A and B). In addition to the reduced radial outgrowth, the number of branch points at the angiogenic front was also significantly lower in EVL$^{ΔEC}$ than in control littermates (Fig 4C). Importantly, endothelial-specific EVL deletion was also associated with a significant reduction in tip cell and filopodia numbers at the vascular front and a significant decrease in endothelial cell proliferation (Fig 4D–F). Together, the data indicate that the impaired angiogenesis in EVL-deficient mice is caused by a reduction in endothelial cell sprouting and proliferation.

**Transcriptome analysis of EVL-deficient retinal endothelial cells**

To identify endothelial pathways regulated by EVL, the gene expression profiles of CD31/CD34 double-positive endothelial cells from P5 wild-type and EVL$^{-/-}$ mouse retinas were compared by RNA-Seq

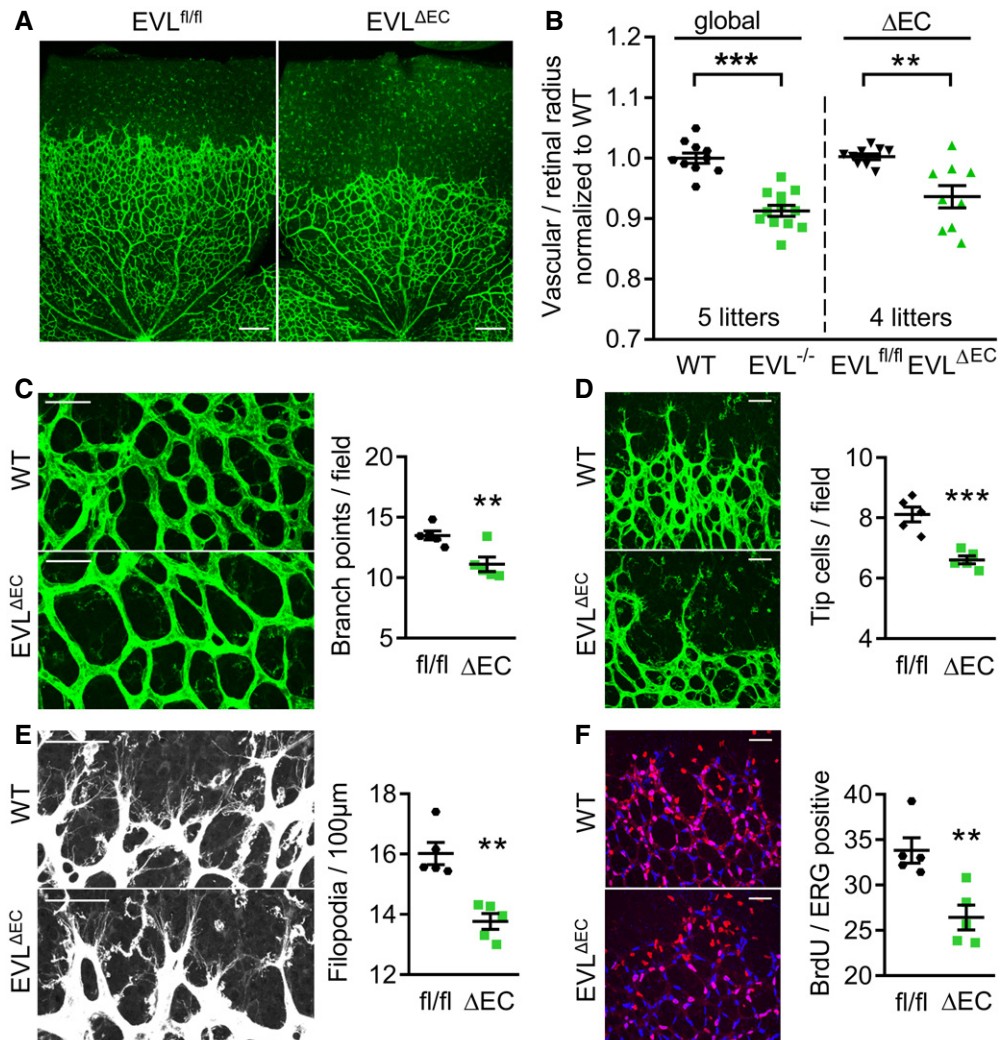

**Figure 4. Impaired retinal angiogenesis in endothelial cell-specific EVL knockout mice.**

The retinal vasculature of endothelial cell-specific EVL knockout mice (EVL$^{\Delta EC}$) and littermate controls (EVL$^{fl/fl}$) on P5 was analyzed by Isolectin B4 staining.

A, B   Radial outgrowth of the retinal vasculature; in (B) global EVL$^{-/-}$ mice are shown for comparison; 4–5 independent experiments.

C        Quantification of the branch points in a field of 170 μm × 300 μm directly behind the vascular front.

D        Tipp cell numbers per field of view (387.5 × 387.5 μm).

E        Filopodia numbers normalized to the length of the angiogenic front.

F        Endothelial cell proliferation analyzed by BrdU incorporation (red) of ERG-positive (blue) endothelial nuclei.

Data information: (C–F) *n* = 5 animals per group from 3 different litters. Scale bars 200 μm (A) and 50 μm (C–F), respectively. Error bars represent SEM; \*\**P* < 0.01, \*\*\**P* < 0.001, unpaired Student's *t*-test with Welch's correction.

(Dataset EV1). As expected, EVL RNA levels were reduced in EVL$^{-/-}$ mice compared with wild-type mice, while those of VASP and Mena were unaffected (Fig 5A). This indicates that EVL deficiency did not elicit the compensatory up-regulation of the other Ena/VASP family members. Gene set enrichment analysis of EVL-deficient vs. wild-type endothelial cells revealed significant changes in angiogenesis-related GO terms (Fig 5B, Dataset EV2). Particularly, the down-regulated gene sets in EVL$^{-/-}$ retinal endothelial cells suggested a function of EVL in blood vessel morphogenesis, endothelium development, and regulation of endothelial cell apoptosis. Furthermore, gene sets associated with regulation of actin filament-based process and regulation of protein transport were also down-regulated in

EVL$^{-/-}$ endothelial cells (Fig 5C, Dataset EV3). Next, we focused on individual dysregulated genes and extracted those genes, which were markedly down-regulated in EVL$^{-/-}$ retinal endothelial cells (log$_2$ fold change < −0.5) and associated with the most significantly changed GO terms "blood vessel development" and "blood vessel morphogenesis" (Table EV1). These genes included cyclooxygenase 2 (Ptgs2), serpine 1, paxillin, and the tip cell marker Esm1, which are well known to promote angiogenic sprouting (Gately & Li, 2004; Rocha *et al*, 2014; Takayama *et al*, 2016; Boscher *et al*, 2019). We confirmed the down-regulation of these genes in EVL-deficient mouse endothelial cells vs. wild-type controls by qPCR (Fig 5D) and mRNA levels of Esm1 and Ptgs2 were also significantly reduced in

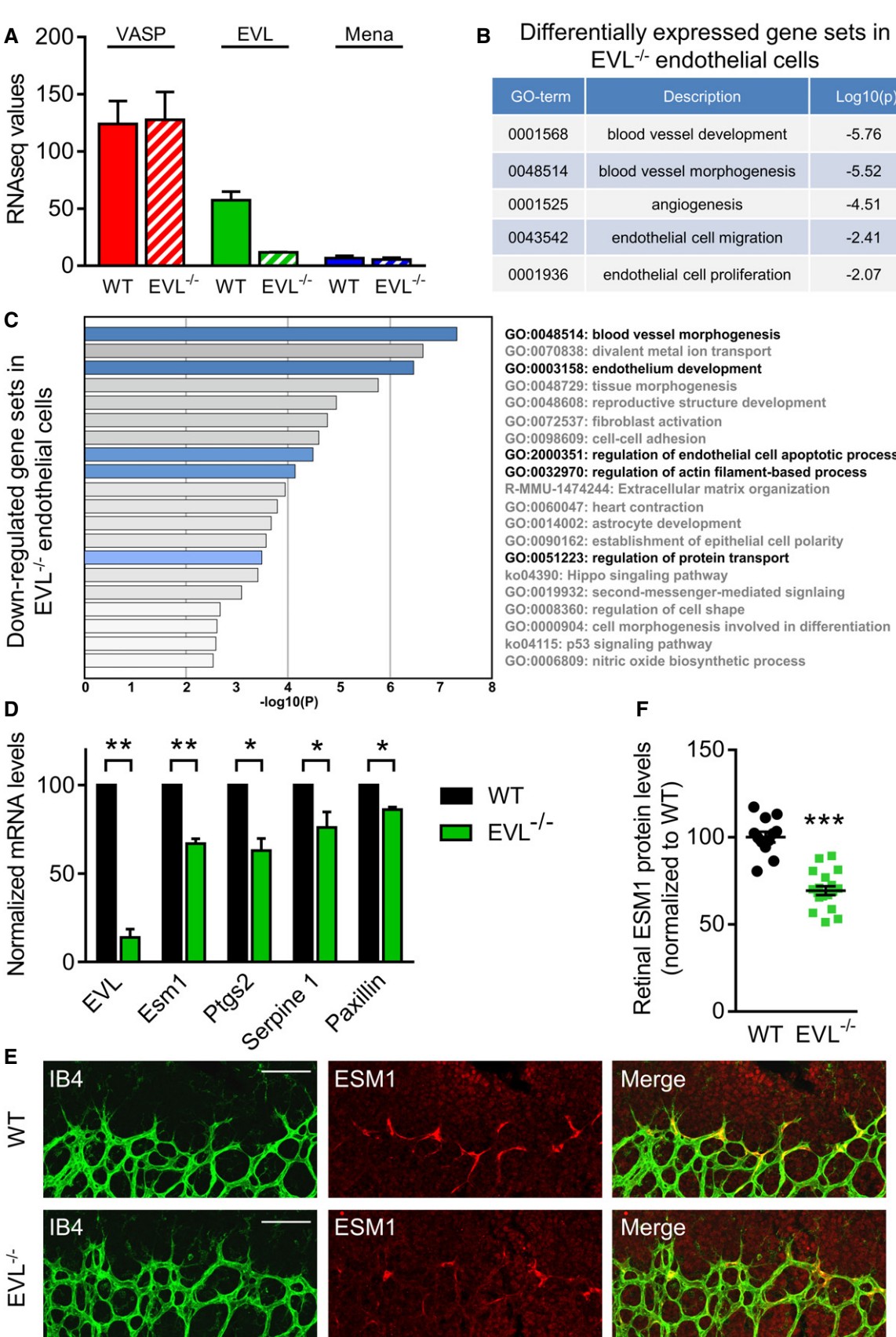

Figure 5.

**Figure 5.   Transcriptome analysis of EVL-deficient retinal endothelial cells.**

CD31 and CD34 double-positive endothelial cells were isolated from wild-type or EVL$^{-/-}$ P5 retinas by FACS and analyzed by RNA sequencing. Two independent experiments; in total 18 retinas from WT mice and 18 retinas from EVL$^{-/-}$ mice from three different litters, each.

A   RNA levels (FPKM) of VASP, EVL, and Mena in endothelial cells from wild-type or EVL$^{-/-}$ retinas.

B   Pathway analysis (metascape.org) was used to identify differentially expressed gene sets in EVL$^{-/-}$ retinal endothelial cells. Selected gene sets involved in vessel branching are shown.

C   Pathway analysis of all under-represented gene sets in EVL$^{-/-}$ endothelial cells.

D   Analysis of mRNA levels of Esm1, Ptgs2 (cyclooxygenase 2), serpine 1 and paxillin in EVL-deficient MLEC relative to WT controls set to 100. $n$ = 3–4 independent experiments, error bars represent SEM, one-sample $t$-test, *$P$ < 0.05; **$P$ < 0.01.

E   ESM1 protein expression in P5 retinas of wild-type (WT) and global EVL$^{-/-}$ mice. Retinas were fixed and stained with isolectin B4 (IB4, green) to visualize endothelial cells and antibodies directed against the tip cell marker ESM1 (red). Representative images from three independent experiments are shown. Scale bars, 100 μm.

F   Analysis of retinal ESM1 protein levels normalized to wild-type littermates. Error bars represent SEM; ***$P$ < 0.001, unpaired Student's $t$-test; three different litters.

human endothelial cells transfected with small interfering RNA directed against EVL (Fig EV4). Consistent with our transcriptome and qPCR data, retinal Esm1 protein levels were significantly reduced in P5 EVL$^{-/-}$ animals (Fig 5E and F), which confirms the reduced number of tip cells at the vascular front of postnatal EVL$^{-/-}$ retinas.

**EVL deficiency impairs VEGF-induced endothelial cell proliferation and sprouting**

Endothelial cell migration/proliferation, tip cell filopodia formation, and blood vessel development are tightly regulated by VEGF and Notch signaling (Pontes-Quero *et al*, 2019). Our transcriptome analysis of retinal endothelial cells (Dataset EV1) revealed no significant differences in DLL4 or Notch expression or expression of Notch downstream targets, e.g., Hey-1/2 or Hes-1. Therefore, we reasoned that impaired VEGF signaling may underlie the compromised postnatal angiogenesis in EVL-deficient mice and analyzed the impact of EVL deficiency on endothelial cell responsiveness to VEGF. Proliferation of wild-type and EVL$^{-/-}$ MLECs was undistinguishable under basal conditions. However, while VEGF stimulation significantly increased the proliferation of wild-type MLECs, VEGF failed to increase the proliferation of EVL-deficient cells (Fig 6A). Interestingly, fibronectin increased the proliferation of both wild-type and EVL$^{-/-}$ cells to the same extent, indicating that the mitotic capacity of EVL-deficient endothelial cells is not generally impaired. This is worth mentioning because similar to growth factor activation, fibronectin-mediated integrin engagement activates multiple growth-associated kinases including ERK1/2 (Wilson *et al*, 2003). Next, the three-dimensional sprouting of wild-type and EVL$^{-/-}$ endothelial cells was assessed in aortic ring assays, a physiologically relevant *in vitro* model for angiogenesis (Baker *et al*, 2012). VEGF-induced endothelial cell sprouting was largely abrogated in EVL$^{-/-}$ aortic rings, whereas no difference was observed under basal conditions (Fig 6B). Besides endothelial cells, other cell types, such as fibroblasts, pericytes, and macrophages, support the sprout formation from aortic rings. To specifically address the contribution of endothelial EVL, we studied sprouting angiogenesis of wild-type or EVL-deficient mouse lung endothelial cells coated on microbeads and embedded into a 3D fibrin gel (Frye *et al*, 2018). While VEGF treatment significantly increased the proportion and sprout length of WT MLEC sprouts, VEGF-induced sprouting was completely blunted in EVL$^{-/-}$ MLEC (Fig 6C–E). Together these findings suggest that EVL regulates VEGF-mediated sprouting angiogenesis *in vitro* and *ex vivo*.

**EVL deficiency impairs VEGFR2 internalization and signaling**

Next, we focused on identifying the mechanisms underlying defective responsiveness to VEGF in EVL-deficient endothelial cells. VEGFR2 levels were comparable in wild-type and EVL$^{-/-}$ MLEC, indicating that the defect was not in receptor expression *per se* (Fig 7A). VEGFR2 internalization is essential for signaling in the endosomal compartment and for VEGF-induced tip cell filopodia formation (Sakurai *et al*, 2005; Sawamiphak *et al*, 2010; Simons *et al*, 2016). Therefore, the role of EVL in VEGFR2 internalization was assessed using a fluorescence antibody feeding assay, which displays surface or internalized receptors in yellow and green, respectively (Fig 7B). Under basal conditions, no differences in VEGFR2 trafficking were observed between wild-type and EVL$^{-/-}$ MLEC. In contrast, however, VEGFR2 clustering and internalization were significantly impaired in VEGF-stimulated EVL$^{-/-}$ cells compared with wild-type controls (Fig 7B). Delayed VEGFR2 internalization (i.e., retention of VEGFR2 at the membrane) exposes the receptor to the phosphatase-rich environment and thus leads to dephosphorylation of the crucial Tyr1173 residue (Lampugnani *et al*, 2006; Lanahan *et al*, 2010; Sawamiphak *et al*, 2010). Consistent with the blunted VEGFR2 internalization, VEGFR2 phosphorylation at Y1173 was significantly reduced in VEGF-stimulated EVL$^{-/-}$ MLEC but not in the absence of the agonist (Fig 8A). Consistent with the fact that phosphorylation of VEGFR2 on Tyr1173 is required for VEGF-induced activation of ERK1/2, phosphorylation of the kinase was significantly impaired in EVL-deficient endothelial cells (Fig 8B) and at the vascular front of EVL-deficient P5 retinas (Fig 8C and D). Together, our data show a role of endothelial EVL in VEGF receptor 2 internalization and downstream signaling with implications for developmental angiogenesis.

# Discussion

The results of the present study revealed that EVL is a regulator of developmental angiogenesis in the postnatal retina. In the absence of EVL, endothelial tip cell density and filopodia formation at the angiogenic front were reduced, which resulted in impaired vascular sprouting. Mechanistically, EVL deletion was found to disturb the VEGF-induced phosphorylation of VEGFR2, as well as its internalization and the downstream activation of ERK1/2.

Branching morphogenesis in neuronal path finding and angiogenesis share many common features from the morphological to the molecular level (Quaegebeur *et al*, 2011; Fischer *et al*, 2018). Given

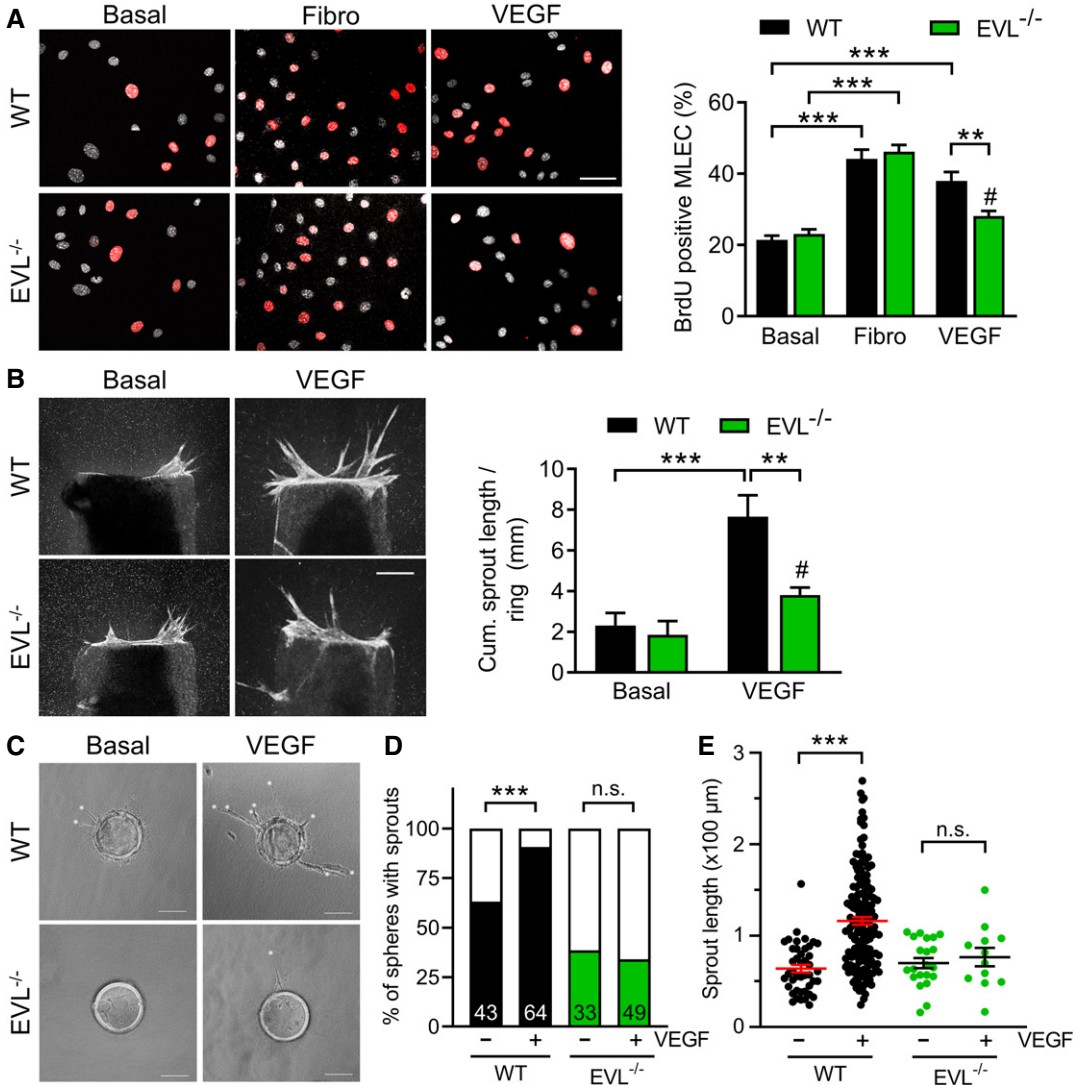

**Figure 6. EVL deficiency impairs VEGF-induced endothelial cell proliferation and sprouting.**

A  Proliferation of WT and EVL$^{-/-}$ MLEC was assessed by BrdU incorporation. The number of BrdU-positive cells (red) was normalized to the total number of cells (white). Eight fields of view per condition and genotype for each of 5 independent WT and EVL$^{-/-}$ cell batches; VEGF stimulation (10 ng/ml) error bars represent SEM; **$P < 0.01$, ***$P < 0.001$, $^{\#}P = 0.171$ non-significant vs. Basal EVL$^{-/-}$, one-way ANOVA with Bonferroni's multi comparison test; scale bar = 50 μm.

B  Cumulative length of CD31-positive sprouts from WT and EVL$^{-/-}$ aortic rings embedded in collagen after 5 days without (basal) or with VEGF stimulation (30 ng/ml). 7 WT and 6 EVL$^{-/-}$ adult animals from three different litters, each; 3 aortic rings per animal and condition; error bars represent SEM; **$P < 0.01$; ***$P < 0.01$; $^{\#}P = 0.543$ non-significant vs. basal EVL$^{-/-}$, one-way ANOVA with Bonferroni's multi comparison test; scale bar, 250 μm.

C  Sprouting of WT and EVL$^{-/-}$ primary MLEC spheres in the absence (Basal) or presence of VEGF (50 ng/ml); asterisks indicate sprouts; scale bars, 100 μm.

D  Solid bars represent proportion of WT and EVL$^{-/-}$ MLEC spheres forming sprouts in the absence (−) or presence (+) of VEGF. n-numbers are indicated; n.s. non-significant ($P = 0.557$); ***$P < 0.001$, Fisher's exact test.

E  Quantification of MLEC sphere sprout length without or with VEGF. Dots represent individual sprouts. Horizontal lines represent mean, error bars represent SEM. $n = 20–30$ beads (from three independent experiments). ***$P < 0.001$, n.s. non-significant ($P = 0.543$), one-way ANOVA with Bonferroni's multi comparison test.

the established role of Ena/VASP proteins in axon outgrowth and guidance, we speculated that the proteins play a role in angiogenic sprouting. We focused on two members of the family, i.e., EVL and VASP, as the expression of Mena was barely detectable in retinal endothelial cells from mouse pups. While both the global and endothelial-specific deletion of EVL significantly compromised sprouting in the retinal vascular plexus, the genetic ablation of VASP did not affect the rate of radial sprouting, even though robust VASP mRNA expression was detected in retinal endothelial cells.

Consistent with this, radial sprouting in retinas from EVL/VASP-double deficient animals was similar to that observed in retinas from EVL-deficient mice. Thus, the role for EVL in retina angiogenesis cannot be compensated for by one of the other family members, as was the case in *Drosophila* Ena$^{-/-}$ mutants (Ahern-Djamali *et al*, 1998).

During active angiogenesis, endothelial tip cells form actin-rich filopodia that sense the environment for guidance cues, such as VEGF, and thereby determine the direction of growth. Given the link

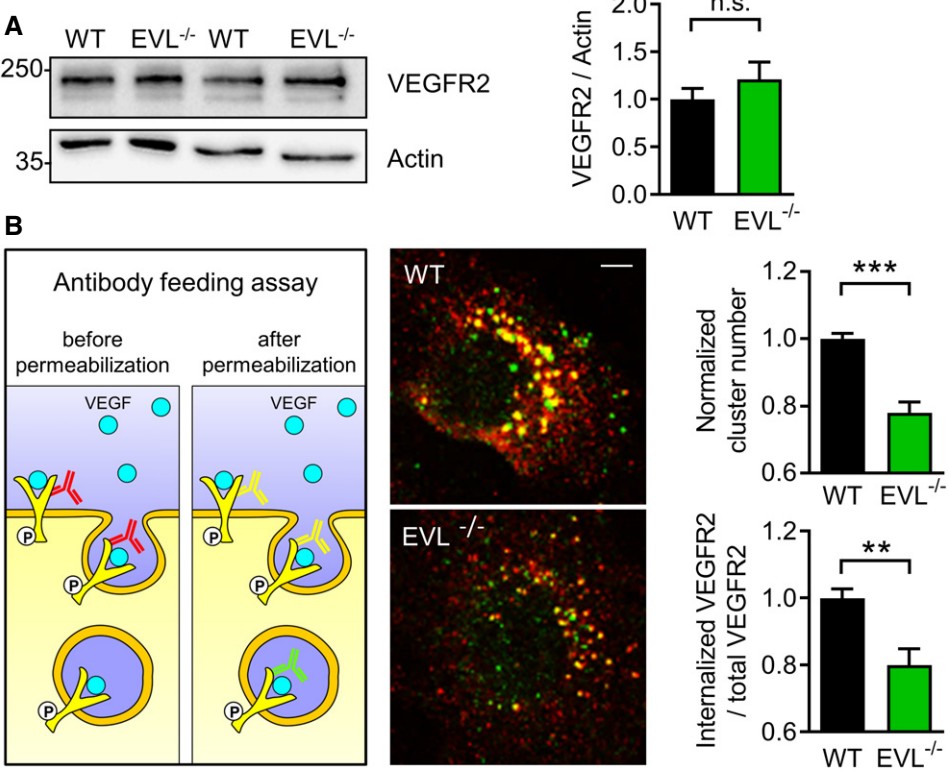

**Figure 7. EVL deficiency impairs VEGFR2 internalization.**

A  VEGFR2 protein levels in wild-type (WT) and EVL-deficient (EVL$^{-/-}$) endothelial cells. Actin was used as loading control. The lower panel shows quantification of VEGFR2 to actin ratios from five independent cell batches; error bars represent SEM; n.s. non-significant ($P = 0.388$), unpaired Student's *t*-test.

B  Antibody feeding assay to monitor VEGFR2 internalization after VEGF stimulation. Left panel: schematic diagram showing the two step staining procedure of surface and internalized VEGFR2 antibodies, which are displayed in yellow and green in the merged confocal images of WT and EVL$^{-/-}$ endothelial cells (middle panel), respectively; scale bar, 10 μm. The bar graphs show quantification of the number of VEGFR2 clusters and the ratio of internalized to total VEGFR2 normalized to wild-type; $n = 4$–6 independent cell batches, error bars represent SEM; **$P < 0.01$, ***$P < 0.001$, unpaired Student's *t*-test.

between actin and the Ena/VASP proteins, it seemed logical to assess their role in tip cell filopodia formation at the angiogenic front. As a consequence of the strong expression of EVL in other cell types of the retina, it was difficult to precisely define the subcellular localization of EVL in endothelial tip cells. However, in migrating endothelial cells *in vitro*, EVL was concentrated at the tips of filopodia- and lamellipodia-like structures and in the absence of EVL filopodia numbers at the vascular front were significantly reduced. These findings add to the *in vitro* evidence that implicates Ena/VASP proteins in filopodia turnover and protrusion persistence in multiple cell types (Schirenbeck *et al*, 2006; Applewhite *et al*, 2007; Drees & Gertler, 2008; Barzik *et al*, 2014; Winkelman *et al*, 2014). Fitting with the localization of Ena/VASP proteins in axonal growth cones, the expression of GFP-VASP fusion protein was detected at the tips of filopodia-like protrusions in sprouting endothelial tip cells during intersegmental vessel development in the zebrafish (Fischer *et al*, 2018). In the absence of filopodia, zebrafish endothelial cells continued to migrate, albeit at reduced velocity, likely because endothelial cells generate compensatory lamellipodia that are sufficient to drive endothelial cell migration when filopodia formation is inhibited (Phng *et al*, 2013). Our own attempts to study sprouting angiogenesis in the developing vasculature of EVL-deficient zebrafish embryos were complicated by the fact that neither TALEN nor

extensive CRISPR/Cas9 genome editing approaches to invalidate the zebrafish EVL gene were successful, probably consecutive to EVL gene methylation.

Alternative splicing generates two EVL isoforms, the standard EVL and a 21 amino acids longer EVL-I variant. Interestingly, EVL-I was only expressed in the postnatal, but not in the adult retina. The *in vivo* function and regulation of EVL-I is still poorly understood but the 21 amino acid insert in EVL-I has been reported to be phosphorylated by protein kinase D (PKD), which may translate PKD activity into filopodia dynamics. Indeed, PKD phosphorylated EVL-I could support filopodia formation and the phosphorylated EVL-I is localized at filopodia tips of migrating cells (Janssens *et al*, 2009). This is noteworthy because PKD has been implicated in endothelial cell migration and angiogenesis, and can be activated by a multitude of vasoactive peptides. Importantly, VEGF induces PKD phosphorylation and activation and increasing evidence suggests a critical role of PKD in VEGF-induced endothelial cell migration and angiogenesis (Wong & Jin, 2005; Ha & Jin, 2009). Thus, it is tempting to speculate that EVL-I executes a specialized role in the developing retina, e.g., in developmental angiogenesis in response to PKD activation. However, further studies and more specialized tools, such as EVL-I-specific antibodies and phosphomimetic EVL mutants, are required to address this hypothesis experimentally.

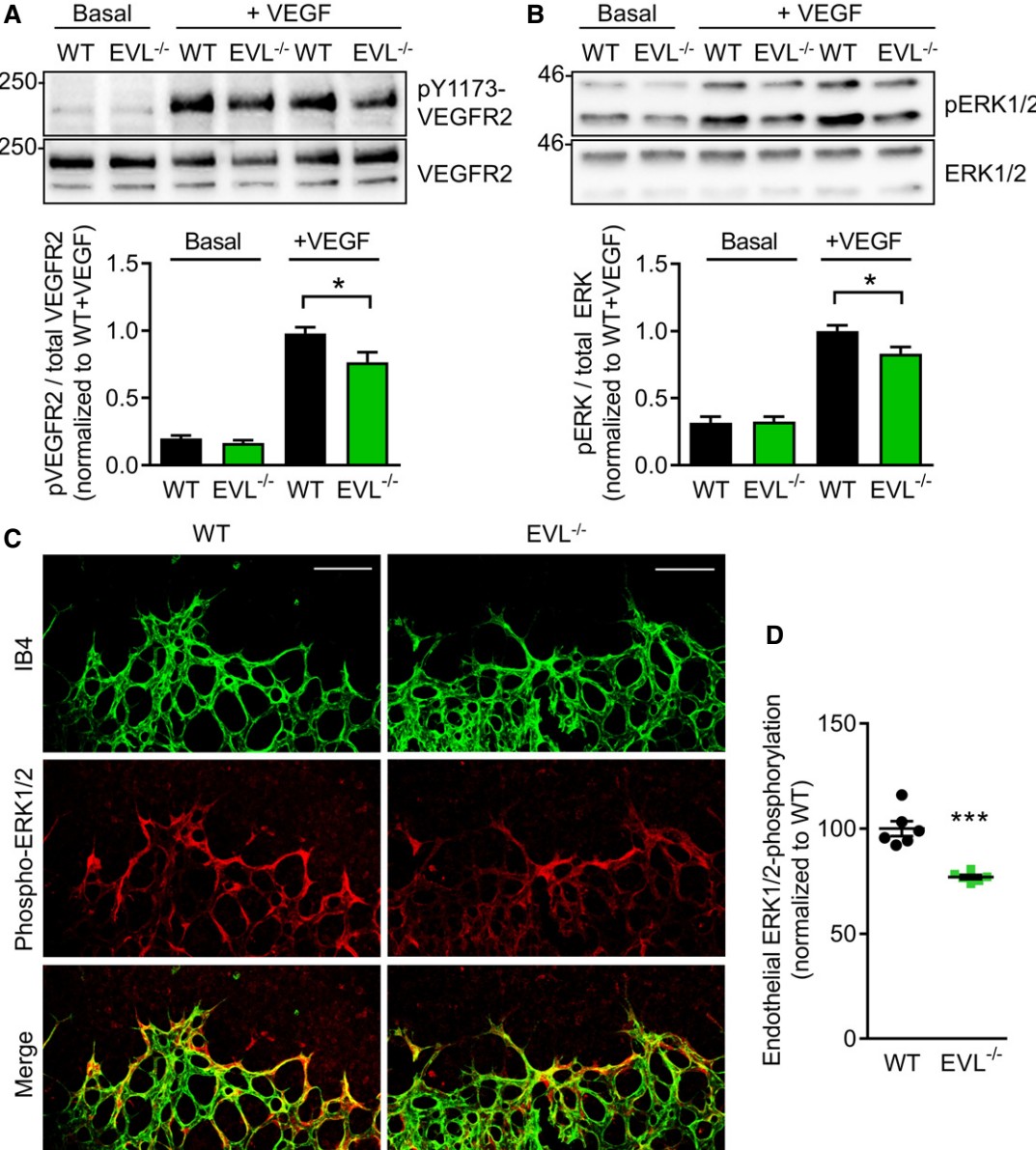

**Figure 8. EVL deficiency impairs VEGFR2 signaling.**

A   Western blots and quantification of VEGFR2 phosphorylation levels relative to total VEGFR2 levels in WT and EVL$^{-/-}$ endothelial cells under basal and VEGF-stimulated conditions (80 ng/ml); $n = 5$ independent cell batches, error bars represent SEM; *$P < 0.05$, one-way ANOVA with Bonferroni's multi comparison test.

B   Western blots and quantification of ERK1/2 phosphorylation levels relative to total ERK levels in WT and EVL$^{-/-}$ endothelial cells under basal and VEGF-stimulated conditions (80 ng/ml); $n = 6$ independent cell batches, error bars represent SEM; *$P < 0.05$, one-way ANOVA with Bonferroni's multi comparison test.

C   ERK1/2 phosphorylation in P5 retinas of wild-type (WT) and global EVL$^{-/-}$ mice. Retinas were fixed and stained with isolectin B4 (IB4, green) to visualize endothelial cells and antibodies directed against phospho-ERK1/2 (red). Representative images from three independent experiments are shown. Scale bars, 100 μm.

D   Analysis of endothelial ERK1/2 phosphorylation normalized to wild-type littermates. Error bars represent SEM; ***$P < 0.001$, unpaired Student's t-test; two different litters.

The present study indicates a role of EVL beyond filopodia formation. Indeed, the *in vivo* deficiency of EVL reduced endothelial cell proliferation as well as the number of tip cells at the vascular front. RNA sequencing of retinal endothelial cells on P5 revealed marked changes in the transcriptome of EVL-deficient cells, and many of the genes affected were associated with impaired endothelial cell function and blood vessel development. The changes observed were similar to those expected as a result of defects in VEGF receptor 2 internalization and signaling. Indeed, endothelial cell proliferation and sprouting in response to VEGF stimulation were blunted in the absence of EVL, even though EVL$^{-/-}$ cells responded normally to fibronectin. Given that fibronectin-mediated proliferation involves ERK phosphorylation, which is also a key driver of VEGF-induced endothelial proliferation, this suggests that

ERK1/2 signaling is likely not impaired in EVL-deficient endothelial cells *per se*. While VEGFR2 levels were not affected by the lack of EVL, the VEGF-induced phosphorylation of the receptor and its internalization and downstream ERK1/2 activation were all impaired in the absence of EVL. VEGF receptor phosphorylation and internalization are crucial for tip cell filopodia extension and vascular sprouting (Sawamiphak *et al*, 2010; Wang *et al*, 2010; Simons *et al*, 2016); however, downstream mechanisms and proteins involved in VEGF receptor endocytosis and signaling are not well defined. Given our previous observation that VASP regulates leukocyte responsiveness and chemotaxis at least partially via altered CCR2 chemokine receptor trafficking (Laban *et al*, 2018), the role of EVL in VEGFR2 internalization was investigated. Similar to the reduced CCR2 receptor internalization in VASP-deficient leukocytes, EVL deficiency significantly impaired the internalization and phosphorylation of the VEGFR2 and its downstream signaling via the MAPK/ERK pathway in endothelial cells. Given that the magnitude of hypoxia/VEGF release decreases in the postnatal mouse retina from P0 to P7 (West *et al*, 2005; Ruiz de Almodovar *et al*, 2009), it seems logical that the impact of EVL deficiency on endothelial proliferation and sprouting may be more prominent at earlier points in time, e.g., P3/P5 vs. P7. Indeed, retinal endothelial cell proliferation at P7 was undistinguishable between wild-type and EVL-deficient animals (Fig EV5C).

The regulation of receptor internalization seems to be a role common to the Ena/VASP family members as the silencing of also Mena impaired the clathrin-mediated endocytosis of the EGF receptor in HeLa cells, which may account for its role in EGF-dependent breast cancer invasion and metastasis (Philippar *et al*, 2008; Vehlow *et al*, 2013). Notably, VEGFR2 membrane trafficking conforms, at least in some respect, that of EGFR (Horowitz & Seerapu, 2012; Zhang & Simons, 2014). How exactly Ena/VASP proteins regulate receptor trafficking and which receptor classes are affected remains to be determined but it is tempting to speculate on the regulation of actin dynamics. Regulation of actin dynamics has a central role in processes that reshape the plasma membrane. This is not limited to protrusions of lamellipodia and filopodia during cell migration but also includes different forms of endocytosis, including phagocytosis, macropinocytosis, and clathrin- or caveolae-mediated endocytosis. Given that actin dynamics are involved in multiple steps of endocytosis, including formation of membrane invaginations, fission of vesicles from the plasma membrane and vesicle movement, actin networks must be tightly regulated for efficient internalization (Kaksonen *et al*, 2006; Mooren *et al*, 2012). A recent study in human endothelial cells has reported that VEGF-induced internalization and signaling of VEGFR2 is largely mediated by macropinocytosis, an actin driven form of endocytosis. Upon VEGF stimulation, a VEGFR2-positive vesicle formed from a ring of actin polymerizing beneath the plasma membrane (Basagiannis *et al*, 2016). So far, Ena/VASP proteins have not been functionally implicated in macrospinocytosis. However, the latter process closely resembles phagocytosis (Bloomfield & Kay, 2016) and recruitment of Ena/VASP proteins to the phagocytic cup is required for the remodeling of the actin cytoskeleton and the efficient particle internalization (Coppolino *et al*, 2001). Furthermore, actin assembly in endosomal "comet tails" seems to push the endosomes along (Kaksonen *et al*, 2006). This is reminiscent of the intracellular bacterial pathogen Listeria, which uses a very similar actin rocketing mechanism for its

motility inside mammalian host cells. This is worth mentioning as the Listeria surface protein ActA has been shown to recruit Ena/VASP proteins to promote local actin polymerization and intracellular motility (Niebuhr *et al*, 1997). Currently, however, it is impossible to rule out that complex formation between EVL and VEGFR2 or regulation of clathrin-mediated endocytosis may regulate receptor internalization in an actin-independent manner. Indeed, Ena/VASP proteins directly interact with several other guidance cue receptors, including Robo, Sema6A, and Dlar (Wills *et al*, 1999; Bashaw *et al*, 2000; Klostermann *et al*, 2000).

Irrespective of the molecular mechanism(s) and the cell types involved, Ena/VASP-dependent receptor internalization seems to occur predominantly in the context of chemotaxis and guidance cue mediated cell migration. This includes leukocyte chemotaxis in response to CCR2 trafficking (Laban *et al*, 2018), breast cancer cell invasion and metastasis in response to EGFR internalization (Philippar *et al*, 2008; Vehlow *et al*, 2013), ephrin/Eph-mediated fibroblast repulsion via Eph receptor internalization (Evans *et al*, 2007) and potentially other situations, in which Ena/VASP proteins were implicated in attractive or repulsive guidance cue signaling (Wills *et al*, 1999; Bashaw *et al*, 2000; Klostermann *et al*, 2000; Lebrand *et al*, 2004; Toyofuku *et al*, 2004).

Deciphering the similarities between axonal growth cones and endothelial tip cells has dramatically increased our knowledge of many guidance cues, receptors and pathways involved in tip cell navigation and vessel guidance. The present study reports a novel role for EVL in endothelial cell migration and angiogenesis and therefore suggests that even effector molecules that translate guidance cues into cytoskeletal dynamics and cell movement are conserved between axonal growth cones and tip cells. However, our study also indicates that particularly EVL plays a role beyond filopodia formation alone, namely in regulating VEGFR2 internalization and signaling at the vascular front.

## Materials and Methods

### Materials

DMEM-F12, fetal calf serum (FCS), penicillin/streptomycin, cell culture grade BSA and Alexa Fluor-conjugated secondary antibodies, and anti-actin probes were from Thermo Fisher Scientific (donkey-anti-rabbit 488 #A21206, donkey-anti-goat 546 #A11056, donkey-anti-mouse 647 #A31571, donkey-anti-rat 594 #A21209, donkey-anti-mouse 647 #A31571, Alexa Fluor 546 and 660 Phalloidin #A22283 and #A22285, respectively).

### Animals

The mouse EVL gene is composed of 14 exons located on chromosome 12 and alternative splicing of exon 11 generates two protein isoforms, the short EVL protein (393 amino acids) or the 21 amino acids longer EVL-I protein (Lambrechts *et al*, 2000). To assess the role of EVL *in vivo* and to discriminate between the cell type-specific effects of the protein, we obtained mutant EVL mice from the European Mouse Mutant Archive (EMMA; EPD0322_2_A03, EMMA ID EM:06361), which are based on the "knockout first" conditional allele (Skarnes *et al*, 2011). This approach combines the advantages

of both a reporter-tagged null allele and a conditional mutation. The initial unmodified allele (EVL$^{-/-}$) generates global knockout mice through splicing to a lacZ trapping element. Conditional alleles (EVL$^{fl/fl}$) were generated by crossing to transgenic FLP-deleter strain (Tg(ACTFLPe)9205Dym/J, a generous gift from Bernhard Nieswandt and David Stegner, Rudolf-Virchow-Zentrum für Experimentelle Biomedizin, Universität Würzburg, Germany), because FLP recombinase activity removes the splice acceptor, lacZ gene, and neomycin resistance cassette, leaving loxP sites flanking the critical EVL exons 4–6 in place. Endothelial-specific deletion of EVL (EVL$^{\Delta EC}$) was mediated by crossing conditional mice to tamoxifen-inducible Pdgf-iCre/ERT2 transgenes (Claxton *et al*, 2008), which generates a frameshift-induced non-sense-mediated decay of all known protein-coding transcripts (Fig 2A). EVL homozygous mutant animals were viable and fertile and macroscopically indistinguishable from their wild-type littermates. The body weights of P3 and P5 wild-type and EVL$^{-/-}$ animals were almost identical (Fig EV5A), indicating that the postnatal development in EVL-deficient mice is not globally impaired. Similarly, body weights of P5 EVL$^{\Delta EC}$ and littermate controls (EVL$^{fl/fl}$) were statistically indistinguishable (Fig EV5B). All animals were bred at the Goethe University Hospital animal facility and housed in conditions that are conform to the Guide for the Care and Use of Laboratory Animals published by the U.S. National Institutes of Health (NIH publication No. 85-23). EVL$^{\Delta EC}$ and EVL$^{fl/fl}$ mice were treated with tamoxifen (2.0 mg/ml, in sunflower oil, 25 μl ip, Abcam, #ab141943) once per day from days 1 to 4. For the isolation of organs, mice were sacrificed using 4% isoflurane in air and subsequent exsanguination or decapitation. All experiments and pre-established no-go criteria were approved by the governmental authorities (Regierungspräsidium Darmstadt, #FU-1111).

**Retina whole mount staining and analysis**

Animals were sacrificed at P3, P5, or P7 and eyeballs fixed in 4% PFA for 1 h. Dissected retinas were equilibrated and permeabilized in PBlec buffer (1 mM CaCl$_2$, 1 mM MgCl$_2$, 0.1 mM MnCl$_2$, and 0.5% Triton X-100 in PBS, pH 6.8) for 15 min and stained with FITC labeled Isolectin B4 (10 μg/ml, Sigma-Aldrich #L2895) in PBlec overnight at 4°C. After washing with PBS, retinas were flat-mounted in Mowiol mounting medium (25% glycerol, 0.1% Mowiol, and 5% DABCO in 0.1 M Tris, pH 8.5) or Dako fluorescence mounting medium and analyzed using a confocal microscope (Leica, SP8). The radial vascular outgrowth was assessed with a 10-fold objective by measuring the radius from the optic nerve to the vascular front and normalizing it to the retinal radius at 12 positions per retina and further normalizing it to the average of littermate controls. Detailed recordings of tip cells, filopodia, branch points, and cell proliferation were performed with a 40-fold oil immersion objective in eight individual 388 μm × 388 μm fields (branch points 170 μm × 300 μm) in comparable regions at the vascular front. All image analyses were performed with Fiji (ImageJ). For quantification of ESM1 (R&D Systems #1999), pERK (essentially as described elsewhere (Pontes-Quero *et al*, 2019), Cell Signaling Technology, mAb #4370), and Ki67 (Abcam #Ab 16667) signals within the retina, we applied a threshold for IB4 staining, created a threshold-based mask and measured pixel intensities per area (mean gray value). 4–9 maximum-projected images (ESM1 600 × 300 μm, pERK

580 × 580 μm, Ki67 620 × 620 μm) were acquired from two to three litters.

**Endothelial cell proliferation *in vivo***

To label proliferating cells in P5 retinas, BrdU (50 mg/kg, ip, Roche, #10280879001) was injected 4 h before sacrifice. Eyeballs were fixed in 4% PFA for 1 h and dissected retinas were rinsed briefly in water and incubated in 2N HCL for 2 h. After three washes with PBS, retinas were stained with Isolectin B4 in PBlec overnight at 4°C. Retinas were blocked and permeabilized in 10% goat serum (GeneTex, # GTX73206) and 0.25% Triton X-100 in PBS for 6 h at 4°C and subsequently incubated with the primary antibodies anti-BrdU (1:50, BD Biosciences, #347580) and anti-ERG (1:500, Abcam, #ab110639) in blocking buffer overnight at 4°C. Secondary detection was performed with Alexa Fluor-coupled secondary antibodies in blocking buffer for 6 h at 4°C. Retinas were flat-mounted and analyzed as described in the previous section. Proliferating BrdU-positive endothelial cells were normalized to the total number of ERG-positive endothelial cells in 388 μm × 388 μm areas at the proliferating front.

**Retina endothelial cell sorting**

Retinas from WT and EVL$^{-/-}$ animals were dissected from P5 mice in ice-cold Hank's Balanced Salt Solution, slightly dissociated by pipetting and digested with collagenase 1 (300 U/ml) and dispase 2 (3.2 U/ml) in DMEM/F12 medium for 8 min at ambient temperature. The tissue was dissociated well by further pipetting and filtered through a 40 μm cell strainer into FCS to stop the digestion. 20 μg/ml DNase was added to prevent clumping. All further steps were performed on ice or at 4°C. Cells were washed with DMEM/12 plus 10% FCS and resuspended in FACS buffer (10% FCS, 1 mM EDTA, and 0.1% NaN3 in PBS). Cells were blocked with CD16/CD32 (Bd Pharmingen, #553142) for 15 min and stained with antibodies against CD31 (PE-Cy7, BioLegend, #102418) and CD34 (FITC, BD Pharmingen, #560238) for 30 min as well as with 7-AAD (BD Pharmingen, #559925) for 10 min. After washing, cells were resuspended in FACS buffer and CD31, CD34-positive, and 7-AAD-negative endothelial cells were sorted with a BD Bioscience FACSAria III Sorter. Sorted cells were directly lysed in TRIzol lysis reagent (Invitrogen, Thermo Fisher Scientific) to proceed with RNA-Seq analysis.

**RNA sequencing analysis**

RNA isolation of sorted endothelial cells was performed according to the standard RNA isolation procedure using TRIzol™ reagent with the addition of Glycogen (0.1 mg/ml, Sigma-Aldrich) before RNA precipitation for a better visualization of the pellet. The RNA from a total of 18 retinas per sample was pooled and RNA-seq libraries were prepared using the TruSeq mRNA Library Prep Kit (Illumina) with minor modifications in ligation and amplification. Libraries were checked for sizing on the Fragment Analyzer (Advanced Analytical), pooled, and sequenced on an Illumina HiSeq 4000 sequencer generating 50 bp single-end reads (*ca.* 30–40 Mio reads/sample). Demultiplexing was done by using bcl2fastq v2.17.1.14. For data analysis, the reads were aligned to the Ensembl mouse reference genome (mm10) using STAR 2.5. The overlaps with

annotated gene loci was counted using Cufflinks 2.2.1. Differential gene expression analysis was performed with Cuffdiff 2.2.1. Candidate genes were selected with a minimal 2× fold change and FDR adjusted $P < 0.05$.

A pathway analysis of the differentially expressed genes was performed first of all with all candidates and subsequently with the up-regulated and down-regulated genes separately (http://metascape.org; Zhou et al, 2019).

### Generation of EVL fusion proteins and polyclonal EVL antibodies

Given the poor quality of several commercial EVL antibodies, we generated a polyclonal anti-EVL antibody using a fusion protein that comprised the entire mouse EVL protein sequence. All plasmids were verified by DNA sequencing. Full-length EVL (accession number NM_001163394) was amplified from a mouse spleen cDNA with the following primers: MmEVL_F1_SalI:5′-aaGTCGACaaatgagtgaacagagtatctgcca-3′ and MmEVL_R1_Xho: ttCTCGAGttacgtggtgctgatcccactt and subcloned into pCR2.1Topo vector (Invitrogen). The SalI-Xho fragment was then subcloned into bacterial expression vectors pET-28a+ (His$_6$-tag, Novagen) and pGEX-6P1 (GST-tag, GE Healthcare) and transformed into Rosetta DE3 (Novagen). For protein expression, 5 ml 2YT medium supplemented with chloramphenicol (40 μg/ml) and carbenicillin (100 μg/ml) for the pGEX-6P1 plasmid or supplemented with chloramphenicol (40 μg/ml) and kanamycin (50 μg/ml) for the pET-28a+ plasmid was inoculated with a single colony of freshly transformed bacteria at 37°C for 2 h, stepwise expanded to 3× 1.2 l and incubated at 37°C until the OD$_{600}$ of 0.6 was reached. The temperature was switched to 16°C and 1 h later at an OD$_{600}$ of ~ 0.8 protein expression was induced by addition of IPTG to a final concentration of 0.1 mM and cultures were incubated for another 16 h at the same temperature before bacteria were harvested by centrifugation and frozen to −20°C.

Murine His$_6$-EVL was purified by standard IMAC. Briefly, bacteria were resuspended in ice-cold lysis buffer (100 mM sodium phosphate pH 8 supplemented with 500 mM NaCl, 10 mM imidazole pH 8, 1 mg/ml lysozyme, and protease inhibitor mix). After sonication and centrifugation, supernatants were incubated with HisPur Cobalt Superflow agarose (Pierce) for 30 min at 4°C. The resin was extensively washed in lysis buffer with 40 mM imidazole. Bound proteins were eluted with stepwise increase of imidazole concentration from 80 to 1,000 mM. Elution fractions were analyzed by SDS–PAGE and Coomassie staining with BSA as standard. Relevant fractions were pooled and concentrated and the buffer was exchanged to PBS using Vivaspin columns (Sartorius) with a 10 kDa molecular weight cutoff.

Murine GST-EVL was purified by standard affinity chromatography. Briefly, bacteria were resuspended in ice-cold PBS supplemented with 1 mg/ml lysozyme and protease inhibitor mix. After sonication and centrifugation, supernatants were incubated with GSH sepharose (GE-Healthcare) for 1.5 h at 4°C. After extensive washing with ice-cold PBS, bound proteins were eluted in PBS containing 20 mM GSH and concentrated as detailed above.

Polyclonal rabbit anti-EVL antibodies were raised against purified full-length mouse His$_6$-EVL protein. Antibodies were immunoselected with the corresponding GST-EVL fusion protein. The specificity of anti-EVL antibodies was confirmed by immunoblotting with lysates from mammalian cells (HEK293) overexpressing EVL, Mena,

or VASP, respectively. HEK293 cells were selected for transfection because of their low endogenous EVL and VASP expression (Benz et al, 2009). Consistent with the reported molecular mass of mouse EVL in Western blots (Benz et al, 2009), affinity-purified anti-EVL antibodies detected a dominant band at ~ 60 kDa in EVL-transfected HEK293 cells, but not in MOCK-transfected controls (CMV-expression plasmid without insert). Despite the sequence homology between EVL, VASP, and Mena, the EVL antibodies did not cross-react with any of the other Ena/VASP proteins (Fig EV2B). To further characterize the EVL antibody, the endogenous expression of EVL was assessed in primary human dermal lymphatic endothelial cells (HDLEC) and the human endothelium-derived EA.hy926 cells. Given the high expression of EVL in immune cells (Lambrechts et al, 2000), the human monocytic cell line THP-1 was used as a positive control (Fig EV2C).

### Flow cytometric analysis of retinal endothelial cells and HUVEC

Retinas from P5 EVL$^{−/−}$ animals and WT littermates were digested and stained as described in the "Retina endothelial cell sorting" section. After staining with CD31 and CD34 antibodies, cells were fixed with Cytofix/Cytoperm (BD Biosciences) on ice for 10 min. HUVECs were trypsinized, stained with 7-AAD (BD Pharmingen, #559925) for 10 min, washed with PBS, and also fixed with Cytofix/Cytoperm (BD Biosciences) for 10 min on ice. All cells were permeabilized with Perm/Wash (BD Biosciences) and subsequently stained with anti-EVL antibody for 60 min plus an Alexa Fluor-labeled secondary antibody for 30 min in Perm/Wash on ice. After washing, cells were resuspended in FACS buffer and analyzed on a BD FACS Verse flow cytometer. Data analysis was performed using the DIVA software (Version 8.0.1) from Becton Dickinson.

### Aortic ring assay

To investigate the angiogenic potential of WT and EVL$^{−/−}$ mice, the aortic ring assay was performed as described previously (Zippel et al, 2016). Briefly, aortas were dissected from mice, cleaned, cut into 1 mm rings, and embedded in collagen drops in a 48-well plate. The rings were cultivated in microvascular endothelial cell growth medium (PeloBiotech) containing 10 mM L-glutamine and 2% autologous serum, which was obtained from the same animals and pooled from WT and EVL$^{−/−}$. The stimulation was done with 30 ng/ml murine VEGF (PeproTech, #450-32). After 5 days, the rings were fixed with 4% PFA, immunostained with an anti-CD31 antibody (BD Pharmingen, #550274) and analyzed using confocal microscopy (Leica, SP8).

### Immunofluorescence microscopy of cultured endothelial cells

Lung endothelial cells were isolated from adult WT and EVL$^{−/−}$ mice, cultured as described previously with minor changes (Fleming et al, 2005) and randomly tested for mycoplasma contaminations. Lungs were dissected, minced with a scalpel, and digested with 4 U/ml dispase II in Ham's F12 for 1 h at 37°C. Digested tissue was homogenized by pipetting and filtered through a 70 μm cell strainer into Ham's F12 containing 10% FCS. After washing, CD144-positive endothelial cells were isolated from the remaining cells via anti-CD144-coated magnetic beats using MACS MS column (Miltenyi

Biotec). Isolated endothelial cells were seeded on fibronectin-coated culture plates and cultured in DMEM/F12 medium supplemented with 20% FCS, 4–8 µl/ml ECGS-H, 50 U/ml penicillin, and 50 µg/ml streptomycin, 0.4% endothelial cell growth supplement with bovine brain-derived heparin, 0.1 ng/ml epidermal growth factor, 1 µg/ml hydrocortisone, 1 ng/ml basic fibroblast growth factor, 100 U/100 µg/ml P/S, 50 ng/ml amphotericin B, and 2 mmol/l L-glutamine.

Cells were used to investigate the localization of EVL in endothelial cells and to analyze cell proliferation *in vitro* after stimulation with VEGF. Cells were seeded in mouse endothelial cell culture medium mixed 1:1 with basal medium in 8-well µ-Slides (Ibidi). After cell attachment, medium was changed to basal microvascular endothelial cell growth medium (PeloBiotech) containing 2% FCS, BrdU (3 µg/ml) and if applicable 10 ng/ml VEGF. After 16 h, cells were fixed with 4% PFA, incubated in 2N HCL for 30 min, washed intensively with PBS, and blocked/permeabilized in blocking buffer (5% donkey serum and 0.25% Triton X-100 in PBS). Incorporated BrdU was detected using an anti-BrdU antibody (1:50, BD Biosciences, #347580). For EVL detection, cells were blocked/permeabilized in blocking buffer and incubated with the generated anti-EVL antibody. Phospho-Paxillin was detected with a commercially available antibody (Thermo Fisher, #44-722G). For secondary detection, Alexa Fluor-conjugated secondary antibodies were used and cells were co-stained with 0.4 µg/ml DAPI or Alexa Fluor-conjugated phalloidin in PBS. Cells were covered with 50% glycerol in PBS containing 5% DABCO and analyzed using a confocal microscopy (LSM 780, Carl Zeiss). Proliferating cells were counted using Fiji (ImageJ).

## Isolation of mouse brain endothelial cells

Mouse brain microvascular endothelial cells were isolated from adult WT and EVL$^{-/-}$ as described (Gurnik *et al*, 2016). Briefly, isolated brains without cerebellum and olfactory lobes were rolled on a Whatman filter membrane to remove the meninges. After homogenization of the brains in a dounce homogenizer, a digestion with 0.75% collagenase II (Worthington) and buffer A (153 mM NaCl, 5.6 mM KCl, 1.7 mM CaCl$_2$, 1.2 mM MgCl$_2$, 15 mM HEPES, 10 g/l BSA, pH 7.4; 1) was performed in a 1:1:1 ratio for 1 h at 37°C on a rocker. Myelin was removed by resuspending the samples in 25% BSA and centrifugation for 30 min at 4°C. The remaining pellet was further digested with Collagenase, Dispase (Roche), and DNase I (Worthington) in buffer A for 15 min at 37°C. Cells were finally resuspended in MCDB-131 complete medium and seeded on six-well plates that were pre-coated with 150 µg/cm$^2$ collagen I (Corning). For an additional purification of the endothelial cells, the medium was changed after 4 h to puromycin (5 µg/ml) containing medium for a total of 2–3 days, which selects for p-glycoprotein expressing endothelial cells. After additional 24 h, cells were lysed in TBS buffer (50 mM Tris–HCl and 150 mM NaCl, pH 7.4) containing 1% Triton X-100 and protease/phosphatase inhibitor cocktails and analyzed by Western blotting.

## Bead sprouting assay

EVL$^{-/-}$ and WT MLEC were detached with trypsin, and $1 \times 10^6$ cells were incubated with 1500 Cytodex-3 beads (rehydrated in PBS) for 4–6 h in culture medium suspension at 37°C. Endothelial cell-coated beads were subsequently incubated for 24 h in cell culture dishes to achieve efficient coating of the bead. Beads were then embedded in a fibrin gel (3 mg/ml). EVL$^{-/-}$ and WT MLEC were cultured in the absence or presence of 50 ng/ml of VEGF-A (R&D Systems, 293-VE) for 3 days, after which the beads were photographed for quantification of sprout numbers and length.

## VEGFR2 receptor internalization and phosphorylation

The clustering and internalization of WT and EVL$^{-/-}$ mouse lung endothelial cells were investigated as described by (Lampugnani *et al*, 2006; Sawamiphak *et al*, 2010) with minor modifications. Cells were starved in 1% BSA in DMEM/F12 medium for 18 h. Medium was changed to fresh starving medium, after 1 h cells were precooled on ice for 30 min and incubated with 6.7 µg/ml anti-VEGFR2 antibody (BD Pharmingen, #555307) in fresh starving medium for 1 h on ice. Cells were washed twice with cold starving medium and stimulated with 80 ng/ml VEGF (PeproTech, #450-32) for 20 min at 37°C. After fixation with 4% PFA for 20 min, cells were blocked with 5% donkey serum (GeneTex) in PBS and stained with an Alexa Fluor-labeled secondary antibody without permeabilization to label only the external receptors. After washing, cells were permeabilized with blocking buffer containing 0.25% Triton X-100 and stained again with a differently labeled secondary antibody to visualize all internal and external receptors. In addition, cells were stained with Alexa Fluor-conjugated phalloidin. The cells were investigated by confocal microscopy (LSM 780, Carl Zeiss), and the quantification of receptor clustering and internalization was performed with Fiji (ImageJ). Receptor internalization was determined as the ratio of the difference between total number and external receptors to the total number of receptors.

To assess VEGFR2 phosphorylation (pThr1175) and ERK1/2 phosphorylation (pThr 202/204, 185/187) following receptor internalization, endothelial cells were starved for 5 h with 0.1% BSA in DMEM/F12 and stimulated with 80 ng/ml VEGF for 10 min at 37°C. Cells were lysed in 2% SDS in TBS buffer (50 mM Tris–HCL pH 7.4, 150 mM NaCl and protease/phosphatase inhibitor cocktails) for the detection of VEGFR2 phosphorylation or lysed on ice in 1% Triton X-100 buffer (50 mM Tris–HCl, pH 7.4 and protease/phosphatase inhibitor cocktails) for the detection of ERK1/2 phosphorylation and analyzed by Western blotting.

## Immunoblotting

Cells and mouse organs were lysed in either TBS buffer (50 mM Tris–HCl, 150 mM NaCl, pH 7.4) containing 1% Triton X-100 or 2% SDS and protease/phosphatase inhibitor cocktails or in lysis buffer (20 mM imidazole pH 6.8, 100 mM NaCl, 2 mM CaCl$_2$, 1% Triton X-100, 0.04% NaN$_3$, and 1× complete EDTA free protease inhibitors (Roche)). Organs were homogenized using an Ultra-Turrax (Polytron PT 3100, Kinematica AG, Lucerne), and samples were cleared by centrifugation. Lysates were separated by 8–10% SDS–PAGE, blotted, and blocked either in 3% BSA or 5% dry milk in TBS containing 0.3% Tween-20. Membranes were incubated in primary antibodies overnight at 4°C (EVL (in dry milk), VASP and Mena-specific antibodies were generated as previously described (Benz *et al*, 2013), VEGFR2 (Cell signaling, #2479), pThr1175 VEGFR2 (Cell signaling, #2478), ERK 1/2 (p44/42 MAPK) (Cell signaling,

#9102), pThr 202/204, pThr 185/187 ERK1/2 (Cell signaling, #4376) and β-actin (Cell Signaling Technology, #4970) and proteins were visualized by peroxidase-conjugated secondary antibodies (Merck Millipore, anti-Ms #40125 and anti-Rb #401393) using enhanced chemiluminescence on a Fusion Solo imager (VILBER LOURMAT).

### qPCR analyses

For qRT–PCR analysis of murine lung ECs isolated from EVL-deficient mice and littermate controls, total RNA was isolated by RNeasy Mini kit (QIAGEN) and 0.5 μg was reverse transcribed using SuperScript® VILO cDNA Synthesis Kit (Invitrogen) according to the manufacturer's instructions. Gene expression levels were analyzed using TaqMan Gene Expression Assay (Applied Biosystems) and a 7900 HT Fast Real-Time PCR System thermocycler (Applied Biosystems) following manufacturer's instructions. Relative gene expression levels were normalized to GAPDH. The following probes were used: Hs99999905 GAPDH, Hs00153133 PTGS2, Hs00199831 ESM1, Hs00275226 EVL, Ms99999915 Gapdh, Mm00478374 Ptgs2, Mm00469953 Esm1, and Mm00468405 EVL. qPCR analysis of serpine and paxillin was performed with ABsolute SYBR green master mix (Thermo Fisher Scientific) and gene-specific intron spanning primers in a magnetic induction (MIC) PCR system (BioMolecular Systems). The following primers were used: Paxillin forward, 5′-ctactactgcaacggaccca-3′, Paxillin reverse, 5′-accaaagaaggctccacact-3′; 18S forward, 5′ctttggtcgctcgctcctc-3′, 18S reverse, 5′-ctgaccgggttg-gtttttgat-3′; Serpine, QuantiTect Primer Assay QT00154756 (Qiagen); and EVL, QuantiTect Primer Assay QT00021294 (Qiagen). Human umbilical vein endothelial cells (HUVECs) were isolated and cultured as described (Busse & Lamontagne, 1991). Cells were transfected with EVL siRNA (Qiagen, #SI03105333 pooled with #SI05124903) or control siRNA (Qiagen, #SI03650318) at 90% confluency (60 pmol siRNA per 3.5 cm dish with Lipofectamine RNAiMAX, Thermo Fisher Scientific) and 72–96 h later subjected to qPCR analysis with human probes mentioned above.

### Statistics

GraphPad Prism was used for graphic representation and statistical analysis of the data. Data are expressed as mean ± SEM, and statistical evaluation was performed using Student's $t$-test for unpaired data, one-way ANOVA with Bonferroni's multi comparison test, one-sample $t$-test to compare sample mean with a normalized control value = 100 (Figs 5D and EV5A), and Fisher's exact test to determine association between two categorical variables (Fig 6D). Values of $P < 0.05$ were considered statistically significant. Data distribution was assumed to be normal, but this was not formally tested. The experiments were not randomized. No blinding was done in the analysis and quantifications.

### Extended view material

Figure EV1 shows an overview and the gating strategy of retinal endothelial cell isolation for RNA sequencing. Figure EV2 shows the strategy and verification for the targeted disruption of the mouse EVL gene. Figure EV3 shows EVL localization to focal adhesions in endothelial cells and the postnatal angiogenesis in VASP$^{-/-}$ mice. Figure EV4 shows the mRNA analysis of EVL siRNA transfected HUVEC and the protein knockdown in these cells. Figure EV5 shows the body weights for the wild-type and EVL-deficient mice used in this study and the Ki67 expression in P7 retinas of wild-type (WT) and global EVL$^{-/-}$ mice.

Dataset EV1 shows the transcriptomes of CD31/CD34 double-positive endothelial cells from P5 wild-type and EVL$^{-/-}$ mouse retinas. Dataset EV2 shows the gene set enrichment analysis (metascape.org) of all genes differentially regulated in EVL$^{-/-}$ endothelial cells vs. wild-type controls. Dataset EV3 shows the gene set enrichment analysis of all down-regulated genes in EVL-deficient endothelial cells. Table EV1 shows the genes, which were down-regulated by at least two-fold in EVL$^{-/-}$ retinal endothelial cells vs. wild-type controls and which were associated with the most significantly changed GO terms "blood vessel development" and "blood vessel morphogenesis".

## Data availability

RNA-Seq data produced in this study are available at the Sequence Read Archive (SRA) under the accession number PRJNA610177 at https://www.ncbi.nlm.nih.gov/sra/PRJNA610177.

**Expanded View** for this article is available online.

## Acknowledgements

This work was supported by the Deutsche Forschungsgemeinschaft (SFB 834/A8 to PMB, SFB 834/A5 to IF, SFB 1039/B6 to AW, SFB 841/B8, and SFB 877/A11 to TR). PMB was also supported by the German Center for Cardiovascular Research (DZHK B14-028 SE). MF was supported by the European Union's Horizon 2020 research and innovation program under the Marie Sklodowska-Curie grant agreement No 840189 and the Werner-Otto-Stiftung Hamburg (8/95). The authors are indebted to Isabel Winter, Mechtild Piepenbrock-Gyamfi, Katharina Engel-Herbig, and Praveen Mathoor for expert technical assistance (all Johann Wolfgang Goethe University, Frankfurt, Germany). The authors acknowledge Kavi Devraj (Goethe University Frankfurt, Germany) for help with the isolation of brain endothelial cells, Michael Potente (MPI Bad Nauheim, Germany) for scientific input, Marcus Fruttinger (University College, London, UK) for providing pdgf-driven Cre mice, and Bernhard Nieswandt (University of Würzburg, Germany) for providing transgenic FLP-deleter mice. Open Access funding enabled and organized by ProjektDEAL.

## Author contributions

Research design, experiments, data analysis, figures, and manuscript writing: JZ and MF; Research design, experiments, and/or data analysis: TF, AW, DJ, GS, and RP; Experiments and data analysis: CC, DS, HSt, HL, and LG; Design and project: JH, TR, BV, HSc, and AA-P. Research design, project supervision, and manuscript writing: IF; Conception and study plan, project supervision, research design, experiments, figures, and manuscript writing: PMB.

## Conflict of interest

The authors declare that they have no conflict of interest.

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
