## [Review Process File · EMBO Reports]

EVL regulates VEGF receptor-2 internalization and signaling in developmental angiogenesis

Joana Zink, Maike Frye, Timo Frömel, Claudia Carlantoni, David John, Danny Schreier, Andreas Weigert, Hebatullah Laban, Gabriela Salinas, Heike Stingl, Lea Günther, Rüdiger Popp, Jiong Hu, Benoit Vanhollebeke, Hannes Schmidt, Amparo Acker-Palmer, Thomas Renné, Ingrid Fleming, and Peter Benz

DOI: [10.15252/embr.201948961](https://doi.org/10.15252/embr.201948961)

Corresponding author(s): Peter Benz (benz@vrc.uni-frankfurt.de), Maike Frye (m.frye@uke.de)

Review Timeline:

Submission Date:	28th Jul 19
Editorial Decision:	27th Aug 19
Revision Received:	27th Feb 20
Editorial Decision:	5th May 20
Revision Received:	18th May 20
Editorial Decision:	18th May 20
Revision Received:	16th Oct 20
Editorial Decision:	13th Nov 20
Revision Received:	3rd Dec 20
Accepted:	9th Dec 20

Editor: Deniz Senyilmaz Tiebe

Transaction Report:

Dear Dr. Benz,

Thank you for submitting your manuscript for consideration by EMBO Reports. Three referees agreed to review your manuscript. So far, we have received two referee reports that are copied below. Given that both referees are in fair agreement that you should be given a chance to revise the manuscript, I would like to ask you to begin revising your study along the lines suggested by the referees.

Please note that this is a preliminary decision made in the interest of time, and that it is subject to change should the third referee offer very strong and convincing reasons for this. As soon as/if we receive the final report on your manuscript, we will forward it to you as well.

As you can see, the referees express interest in the proposed role of EVL in sprouting angiogenesis. However, they also raise a number of concerns that need to be addressed to consider publication here.

Given these constructive comments, we would like to invite you to revise your manuscript with the understanding that the referee concerns (as detailed above and in their reports) must be fully addressed and their suggestions taken on board. Please address all referee concerns in a complete point-by-point response. Acceptance of the manuscript will depend on a positive outcome of a second round of review. It is EMBO reports policy to allow a single round of revision only and acceptance or rejection of the manuscript will therefore depend on the completeness of your responses included in the next, final version of the manuscript.

Revised manuscripts should be submitted within three months of a request for revision; they will otherwise be treated as new submissions. Please contact us if a 3-months time frame is not sufficient for the revisions so that we can discuss the revisions further. As a matter of policy, competing manuscripts published during this period will not negatively impact on our assessment of the conceptual advance presented by your study. However, we request that you contact the editor as soon as possible upon publication of any related work, to discuss how to proceed.

Supplementary/additional data: The Expanded View format, which will be displayed in the main HTML of the paper in a collapsible format, has replaced the Supplementary information. You can submit up to 5 images as Expanded View. Please follow the nomenclature Figure EV1, Figure EV2 etc. The figure legend for these should be included in the main manuscript document file in a section called Expanded View Figure Legends after the main Figure Legends section. Additional Supplementary material should be supplied as a single pdf labeled Appendix. The Appendix includes a table of content on the first page with page numbers, all figures and their legends. Please follow the nomenclature Appendix Figure Sx throughout the text and also label the figures according to this nomenclature. For more details please refer to our guide to authors.

- 1) a .docx formatted version of the manuscript text (including legends for main figures, EV figures and tables). Please make sure that the changes are highlighted to be clearly visible.
- 2) individual production quality figure files as .eps, .tif, .jpg (one file per figure).

3) a .docx formatted letter INCLUDING the reviewers' reports and your detailed point-by-point responses to their comments. As part of the EMBO Press transparent editorial process, the point-by-point response is part of the Review Process File (RPF), which will be published alongside your paper. For more details on our Transparent Editorial Process, please visit our website: <https://www.embopress.org/page/journal/14693178/authorguide#transparentprocess>

4) a complete author checklist, which you can download from our author guidelines (<http://embor.embopress.org/authorguide>). Please insert information in the checklist that is also reflected in the manuscript. The completed author checklist will also be part of the RPF.

5) Please note that all corresponding authors are required to supply an ORCID ID for their name upon submission of a revised manuscript (<https://orcid.org/>). Please find instructions on how to link your ORCID ID to your account in our manuscript tracking system in our Author guidelines (<http://embor.embopress.org/authorguide>).

6) We replaced Supplementary Information with Expanded View (EV) Figures and Tables that are collapsible/expandable online. A maximum of 5 EV Figures can be typeset. EV Figures should be cited as 'Figure EV1, Figure EV2' etc... in the text and their respective legends should be included in the main text after the legends of regular figures.

- For the figures that you do NOT wish to display as Expanded View figures, they should be bundled together with their legends in a single PDF file called *Appendix*, which should start with a short Table of Content. Appendix figures should be referred to in the main text as: "Appendix Figure S1, Appendix Figure S2" etc. See detailed instructions regarding expanded view here: <http://embor.embopress.org/authorguide#expandedview>.

7) We would also encourage you to include the source data for figure panels that show essential data.

Numerical data should be provided as individual .xls or .csv files (including a tab describing the data). For blots or microscopy, uncropped images should be submitted (using a zip archive if multiple images need to be supplied for one panel). Additional information on source data and instruction on how to label the files are available <http://embor.embopress.org/authorguide#sourcedata>.

8) Our journal encourages inclusion of *data citations in the reference list* to directly cite datasets that were re-used and obtained from public databases. Data citations in the article text are distinct from normal bibliographical citations and should directly link to the database records from which the data can be accessed. In the main text, data citations are formatted as follows: "Data ref: Smith et al, 2001" or "Data ref: NCBI Sequence Read Archive PRJNA342805, 2017". In the Reference list, data citations must be labeled with "[DATASET]". A data reference must provide the database

name, accession number/identifiers and a resolvable link to the landing page from which the data can be accessed at the end of the reference. Further instructions are available at <http://embor.embopress.org/authorguide#datacitation>.

9) Before submitting your revision, primary datasets (and computer code, where appropriate) produced in this study need to be deposited in an appropriate public database (see <http://embor.embopress.org/authorguide#dataavailability>).

The accession numbers and database should be listed in a formal "Data Availability " section (placed after Materials & Method) that follows the model below. Please note that the Data Availability Section is restricted to new primary data that are part of this study.

Data availability

10) Regarding data quantification, please ensure to specify the name of the statistical test used to generate error bars and P values, the number (n) of independent experiments underlying each data point (not replicate measures of one sample), and the test used to calculate p-values in each figure legend. Discussion of statistical methodology can be reported in the materials and methods section, but figure legends should contain a basic description of n, P and the test applied. Please note that error bars and statistical comparisons may only be applied to data obtained from at least three independent biological replicates. Please also include scale bars in all microscopy images.

I look forward to seeing a revised version of your manuscript when it is ready. Please let me know if you have questions or comments regarding the revision.

Yours sincerely,

Deniz Senyilmaz Tiebe

Deniz Senyilmaz Tiebe, PhD
Editor
EMBO Reports

Referee #1:

Zink et al describe the analysis of Ena/VASP family function in endothelial tip cells using the mouse retina as a model system. They report from RNAseq analysis that only VASP and EVL but not Mena is expressed in P5 mouse retinal endothelial cells. The authors find that EVL but not VASP KO mice show delayed postnatal retinal angiogenesis. The authors use RNAseq and found downregulated pathways associated with angiogenesis, endothelium development and actin filament based processes. Further, they report a reduction in VEGF induced proliferation, angiogenic sprouting, VEGFR2 uptake and VEGFR2 and ERK phosphorylation.

In general this is a very well conducted study and a pleasure to read. It is emerging that the three Ena/VASP proteins fulfil independent functions (for a long time they have been judged as interchangeable). This elegant study provides strong evidence for a specific role of EVL in angiogenesis. Thus I would be happy to support publication in EMBO reports once my major concerns have been addressed.

Major concerns:

It is standard in the field to repeat each experiment at least 3 times. In several experiments this was not the case or it was not reported. Furthermore, in several cases the statistical analysis was flawed.

Specific major concerns:

Fig. 1 B, D. This was repeated only twice. I appreciate that this is an expensive experiment but it is standard in the field to repeat each experiment at least 3 times and a statistical analysis of the data is missing. This is important since in the rest of the study the authors focus on the role of VASP and EVL and not follow up on Mena.

Fig S2B: Which mammalian cells were used for transfection? What is MOCK? No plasmid? Why do you not detect endogenous EVL in these mammalian cells? It would be helpful to show a western blot of a panel of cell lines for further characterisation of this novel anti-EVL antibody.

Fig S2C and page 5: But EVL-I is not expressed in WT brain and retina. This is interesting and could be followed up.

Fig. 2A: This was based on only two KO animals. I don't understand how you can do statistics on this?! It needs to be done with at least three WT and KO animals each.

Fig. 2B,C: How often was this repeated? It needs to be stated in the figure legend that this was repeated at least 3 times. Since you observe expression of EVL and EVL-I and comparing this to Fig S2 C in which you find EVL-I only in P5 but not adult animals, the WB in Fig. 2B,C are from P5 animals? This should be stated here and is very interesting and could be followed up.

Fig. 3B: At P7 only two litters were used. It is standard in the field to repeat each experiment at least 3 times and thus this needs to be based on at least three litters per time point. The individual mice here shown here can be seen as technical repeats and this is used to allow statistical analysis, which is not appropriate.

Fig. 3C: It is standard in the field to repeat each experiment at least 3 times and thus this needs to be based on at least three litters per time point. It needs to be specified in the figure legend on how many litters this experiment was based.

Fig. S3B: At P3 only 2-3 animals from N2 experiments were used. It is standard in the field to repeat each experiment at least 3 times and thus this needs to be based on at least three litters per time point. Since there is a trend for a reduction at P3 for VASP KO mice visible this is important. Only with three litters a statistical analysis is meaningful. Even if it is not significant, the exact P value should be shown in the figure legend.

Fig. 4A: How can you do stats on N=2?

Fig. 5A: It is not clear what n=24 refers to. Is it 24 cells? This would be a very low number for this

type of analysis and from 5 independent cell batches. Is there really not statistically significant increase after VEGF treatment of EVL ko cells compared to basal? What is the p value? This needs to be reported. Looking at the error bars it look like there is a difference.

Fig. 5B,C: From how many independent experiments? This needs to be reported.

Fig. S4: Please show efficiency of knockdown on the protein level using your excellent EVL antibody.

Fig. 6B,C,D: The normalisation to 1.0 was done in an inappropriate way thereby loosing the error bars for the WT condition. This needs to be normalised by anchoring 1.0 to the mean of WT in PRISM. The problem with your approach of normalisation is that the stats program has to assume that all values for the WT are exactly the same (which they are not) and thus they will have an artificially low SD and therefore it is easy to get significance and your stats are not correct.

Fig. 7C,D: The number of independent experiments needs to be reported.

Minor concerns:

The source of the VASP and Mena antibodies is missing in the M&M.

Fig. 2D: It would be helpful to show just the EVL stain in the magnified view to be able to see the localisation of EVL. I also would suggest to replace the black arrows with some other colour as black is very hard to see.

Referee #3:

Zink et al. investigate the role of EVL during sprouting angiogenesis. They find that global or endothelial cell (EC) specific deletion of EVL delays retinal vascular outgrowth. They further show that EC-specific deletion of EVL diminishes filopodia and EC proliferation, as measured by BrDU incorporation. Mechanistically, the authors provide in vitro evidence that loss of EVL causes a decrease in VEGFR2 internalization and ERK phosphorylation. The data are clearly presented and the manuscript is easy to read and follow. Below are my comments how to improve the manuscript.

1. The authors need to tone down their statement (e.g. the first sentence of the discussion) that EVL is an important regulator of developmental angiogenesis in the postnatal retina. Clearly, only a mild delay of angiogenic outgrowth is visible during a narrow time window. At P7 the retinal outgrowth is similar to wildtype embryos.
2. Does the proliferation defect persist after P5 or is BrDU incorporation normal again at P7?
3. It is not clear whether the delay in outgrowth is due to a general delay in postnatal development. Is the body weight of mutant and wildtype embryos comparable?
4. The data seem rather "spread out". For example, figure 1 has little content, panel A could be moved to supplementary figures and expression of EVL could be moved from supplemental figure to main figure. Same is true for figure 4.
5. The data on EVL expression in supplemental figure 1 only show high magnification views. How about overview images? Is EVL enriched in tip cells?

6. The transcriptome analysis of EVL loss of function (Figure 4) needs to be substantiated. The authors only present a pathway analysis, but do not provide information on individual misregulated genes. Which genes are most highly changed in their expression and do the authors observe differences in the expression of these genes using antibody stainings in the retina? Related to this: The authors find a reduction in tip cell numbers in mutant retinas. They need to substantiate this finding by performing a staining for tip cell markers, e.g. ESM1.

7. The results in cultured cells appear more pronounced compared to the in vivo data. Good pERK antibodies are available for staining retinal ECs. Is pERK changed in mutant retinas? Are there differences between P5 and P7? How do the authors explain apparently normal retinal outgrowth after P7 even though the authors claim that VEGFR2 signaling and ERK phosphorylation are reduced?

8. The EC specific knockout figure should be moved after figure 3, as the authors only analyze tip cell and filopodia numbers in the EC specific EVL knockout mice. As the order is right now, I asked myself why the authors do not analyze filopodia in the EVL knockout mice, since they introduce EVL as an important player in filopodial dynamics (I had to wait to the last figure).

9. The abstract is confusing for non-experts. The authors introduce Ena/VASP proteins followed by stating that EVL and VASP, but not Mena are expressed in ECs. For the non-expert it is not clear how EVL and Mena are related to Ena/VASP from the current wording of the abstract.

10. The authors should mention in the discussion that a study in zebrafish found that EC filopodia are dispensable for EC guidance and that their ablation only weakly affected EC migration (Phng et al., 2013, Development).

Referee #1:

Zink et al describe the analysis of Ena/VASP family function in endothelial tip cells using the mouse retina as a model system. They report from RNAseq analysis that only VASP and EVL but not Mena is expressed in P5 mouse retinal endothelial cells. The authors find that EVL but not VASP KO mice show delayed postnatal retinal angiogenesis. The authors use RNAseq and found downregulated pathways associated with angiogenesis, endothelium development and actin filament based processes. Further, they report a reduction in VEGF induced proliferation, angiogenic sprouting, VEGFR2 uptake and VEGFR2 and ERK phosphorylation.

In general this is a very well conducted study and a pleasure to read. It is emerging that the three Ena/VASP proteins fulfil independent functions (for a long time they have been judged as interchangeable). This elegant study provides strong evidence for a specific role of EVL in angiogenesis. Thus I would be happy to support publication in EMBO reports once my major concerns have been addressed.

Reply: We would like to thank the reviewer for his/her constructive criticism of our manuscript and hope that the additional data together with the changes to the text and the comments listed below satisfactorily address the concerns raised.

Major concerns:

It is standard in the field to repeat each experiment at least 3 times. In several experiments this was not the case or it was not reported. Furthermore, in several cases the statistical analysis was flawed.

Specific major concerns:

Fig. 1 B, D. This was repeated only twice. I appreciate that this is an expensive experiment but it is standard in the field to repeat each experiment at least 3 times and a statistical analysis of the data is missing. This is important since in the rest of the study the authors focus on the role of VASP and EVL and not follow up on Mena.

Reply: The reviewer's comment was of course valid. Therefore, we have repeated this experiment two additional times (each sample was a pool of 9 retinas) and also included a statistical analysis of the Ena/VASP expression data. As suggested by reviewer #3, we have also included the immunofluorescence stainings from supplemental figure 1 into figure 1. The revised dataset clearly supports our conclusion that there is virtually no Mena expression in postnatal mouse retinal endothelial cells.

Fig S2B: Which mammalian cells were used for transfection? What is MOCK? No plasmid? Why do you not detect endogenous EVL in these mammalian cells? It would be helpful to show a western blot of a panel of cell lines for further characterisation of this novel anti-EVL antibody.

Reply: We used HEK293 cells for transfection. Our designated "MOCK" represents transfection of HEK293 cells with the empty CMV-expression plasmid. We have now updated this information in the figure legend of Figure EV2 and in the materials and methods section (page 17). HEK293 cells were selected for EVL transfection because these cells express low/undetectable levels of endogenous EVL and VASP (Benz et al., 2009) and are therefore well suited for the characterization of the new EVL antibody. We agree that this is an important point and added the corresponding information on page 17. As suggested, we further characterized the EVL antibody by additional Western blots with lysates of primary endothelial cells and the EA.hy926 cell line (Figure EV2C and results page 5-6). Interestingly, both EVL isoforms were detected in EA.hy926 cells, while only the short EVL isoform was detected in primary human dermal lymphatic endothelial cells (HDLEC) (Figure EV2C). Given the high expression of EVL in

immune cells (Lambrechts et al., 2000), the human monocytic cell line THP-1 was used as a positive control (Figure EV2C).

Fig S2C and page 5: But EVL-I is not expressed in WT brain and retina. This is interesting and could be followed up.

Reply: We fully agree with the comment of the reviewer. Particularly that EVL-I is expressed in the postnatal, but not in the adult retina is interesting. The *in vivo* function and regulation of EVL-I is still poorly understood. Interestingly, however, a previous publication showed that the 21 amino acid insert in EVL-I is phosphorylated by protein kinase D (PKD), which may translate PKD activity into filopodia dynamics. Indeed, PKD phosphorylated EVL-I could support filopodia formation and the phosphorylated EVL-I is localized at filopodia tips of migrating cells (Janssens et al., 2009). This is noteworthy because PKD has been implicated in endothelial cell migration and angiogenesis, and can be activated by a multitude of vasoactive peptides. Importantly, VEGF induces PKD phosphorylation and activation and increasing evidence suggests a critical role of PKD in VEGF-induced endothelial cell migration and angiogenesis (Ha and Jin, 2009; Wong and Jin, 2005). Thus, it is tempting to speculate that EVL-I executes a specialized role in the developing retina, e.g. in developmental angiogenesis in response to PKD activation. However, further studies and more specialized tools, such as EVL-I specific antibodies and phosphomimetic EVL mutants, are required to experimentally address this hypothesis. We have added this information in the discussion section (page 10).

Fig. 2A: This was based on only two KO animals. I don't understand how you can do statistics on this?! It needs to be done with at least three WT and KO animals each.

Reply: We agree with the reviewer's concern. However, as we had to transfer our EVL^{-/-} mice to a new animal facility and the required embryo transfer failed twice we have not been able to complete additional experiments within the deadline of the revision. As a consequence, we have removed the statistical analysis of the data in the revised version of figure 2A. However, given that we provide additional and independent experimental proof (Western blots, immunofluorescence microscopy, next generation sequencing and qPCR), we still believe that it is valid to report the successful deletion of the EVL mRNA and protein in endothelial cells of EVL^{-/-} animals.

Fig. 2B,C: How often was this repeated? It needs to be stated in the figure legend that this was repeated at least 3 times.

Reply: The Western blots (Figure 2B and C) are representative of three independent experiments. We have now included this information in the figure legend.

Since you observe expression of EVL and EVL-I and comparing this to Fig S2 C in which you find EVL-I only in P5 but not adult animals, the WB in Fig. 2B,C are from P5 animals? This should be stated here and is very interesting and could be followed up.

Reply: The Western blots in Fig. 2B, C were generated with endothelial cells isolated from adult mouse brain or lung, respectively. We have updated this information in the materials and methods section on page 18. However, prior to Western blotting the cells were cultured under sparse/migrating conditions, which may thus reflect the situation in the developing P5 retina. We added this information to the figure legend on page 27. It would indeed be very interesting to further explore the role of EVL-I isoform in migrating endothelial cells and angiogenic vessels. However, as outlined above, further studies and more specialized tools, such as EVL-I specific antibodies and phosphomimetic EVL mutants, are required to experimentally address this hypothesis in more detail.

Fig. 3B: At P7 only two litters were used. It is standard in the field to repeat each experiment at least 3 times and thus this needs to be based on at least three litters per time point. The individual mice here shown here can be seen as technical repeats and this is used to allow statistical analysis, which is not appropriate.

Reply: We apologize for the oversight. The five WT and EVL^{-/-} animals shown at P7 originated from three different litters. The “2” was a leftover from a previous version of the manuscript. To have adequate controls, we always used WT and EVL-KO animals from EVL^{+/-} x EVL^{+/-} breeding pairs, resulting in only 25% WT and 25% KO. We have corrected this mistake in the revised version of the manuscript.

Fig. 3C: It is standard in the field to repeat each experiment at least 3 times and thus this needs to be based on at least three litters per time point. It needs to be specified in the figure legend on how many litters this experiment was based.

Reply: We apologize for the oversight. These data were generated from 5 litters for EVL, 6 litters for VASP and 6 litters for EVL+VASP. We have now included this information in Figure 3C in the same way as for Figure 3B. The incorrect numbers were a copy and paste error from a previous version of the manuscript.

Fig. S3B: At P3 only 2-3 animals from N2 experiments were used. It is standard in the field to repeat each experiment at least 3 times and thus this needs to be based on at least three litters per time point. Since there is a trend for a reduction at P3 for VASP KO mice visible this is important. Only with three litters a statistical analysis is meaningful. Even if it is not significant, the exact P value should be shown in the figure legend.

Reply: As suggested, additional experiments were performed with P3 WT and VASP-KO animals and included in the revised version of the figure. We have also included the exact P values in the figure legend (Figure EV3).

Fig. 4A: How can you do stats on N=2?

Reply: The number 2 in this case is misleading as the data originated from two independent experiments; which represent a total of 18 WT and 18 EVL^{-/-} retinas from three different litters. Moreover, the cells were FACS sorted (CD31 and CD34 double positive) endothelial cells. We chose this experimental design to obtain robust NGS data with low variation. We included Figure 4A in the manuscript to show that EVL mRNA was reduced in the endothelial cells from EVL-KO mice, whereas no compensatory up-regulation of VASP and Mena was observed as has been indicated in other Ena/VASP publications. Although we agree that a statistical analysis of two samples is not accurate, we believe that our conclusion is still valid and we have included additional qPCR data in the revised manuscript, that supports our statement that EVL mRNA is reduced in endothelial cells from EVL^{-/-} animals vs. wild-type controls (Figure 5D). Nevertheless, as suggested by the reviewer, we removed the statistics from Figure 4A (which is now Figure 5A in the revised version of the manuscript) and revised our statement in the results section.

Fig. 5A: It is not clear what n=24 refers to. Is it 24 cells? This would be a very low number for this type of analysis and from 5 independent cell batches. Is there really not statistically significant increase after VEGF treatment of EVL ko cells compared to basal? What is the p value? This needs to be reported. Looking at the error bars it look like there is a difference.

Reply: The data in Figure 5A (now Figure 6A in the revised version of the manuscript) correspond to 5 independent experiments (5 independent cell batches) with 8 “fields of view” per condition and genotype, e.g. 8 images for WT-basal/Fibro/VEGF (n = 3x8 = 24 for each of the five WT and KO cell batches). We have clarified this in the revised figure legend of the

manuscript. There is no significant difference between EVL-KO basal vs. VEGF. The p-value is 0.171. We added this information to the figure legend on page 29.

Fig. 5B,C: From how many independent experiments? This needs to be reported. Fig. S4: Please show efficiency of knockdown on the protein level using your excellent EVL antibody.

Reply: The data originate from 7 WT and 6 EVL^{-/-} adult animals from three different litters, each. We analyzed 3 aortic rings per animal and condition. This information is given in the revised figure legend (page 29, Figure 6 in the revised version of the manuscript) and we also included the exact p-value for EVL-KO Basal vs. VEGF (p=0.543). Moreover, we performed additional experiments and analyzed the efficiency of the siRNA-mediated knockdown of EVL on protein level by FACS and Western blotting. Similar to the mRNA levels, EVL protein depletion was not complete (Figure EV4B) and we assume that the remaining EVL protein levels were sufficient to increase HUVEC sprouting in response to VEGF-treatment (although there was a significant difference between control+VEGF and EVL-siRNA+VEGF). To address the full contribution of EVL to endothelial sprouting, we replaced the HUVEC sprouting assay with the sprouting of wild-type and EVL^{-/-} mouse MLEC coated on cytodex microcarrier beads in response to VEGF treatment. Importantly (and unlike the HUVEC experiment), EVL-deficiency almost completely blocked the VEGF-induced increase in endothelial sprouting (both the proportion of spheres, which displayed sprouts and length of the individual sprouts; revised Figure 6C-E).

Fig. 6B,C,D: The normalisation to 1.0 was done in an inappropriate way thereby losing the error bars for the WT condition. This needs to be normalised by anchoring 1.0 to the mean of WT in PRISM. The problem with your approach of normalisation is that the stats program has to assume that all values for the WT are exactly the same (which they are not) and thus they will have an artificially low SD and therefore it is easy to get significance and your stats are not correct.

Reply: We agree with the reviewer that an unpaired Student t-test with normalized data is not statistically accurate to compare the groups. We have now revised the statistics this Figure (Figure 7 B-D in the revised manuscript) as suggested by the reviewer and included additional experimental data. Similar to the original analyses, VEGFR2 internalization and signaling is significantly different between wild-type and EVL-deficient cells.

Fig. 7C,D: The number of independent experiments needs to be reported.

Reply: The analyzed animals originated from 3 different litters. We have now included this information in the figure legend (Figure 4 C, D in the revised version of the manuscript, page 28).

Minor concerns:

The source of the VASP and Mena antibodies is missing in the M&M.

Reply: We apologize for the missing information and have now added the source of the VASP and Mena-specific antibodies in the M&M section (page 20).

Fig. 2D: It would be helpful to show just the EVL stain in the magnified view to be able to see the localisation of EVL. I also would suggest to replace the black arrows with some other colour as black is very hard to see.

Reply: As suggested, we have now included the magnified view with only the EVL staining. We agree that the black arrows were not easy to see and have adjusted this in the revised figure.

Referee #3:

Zink et al. investigate the role of EVL during sprouting angiogenesis. They find that global or endothelial cell (EC) specific deletion of EVL delays retinal vascular outgrowth. They further show that EC-specific deletion of EVL diminishes filopodia and EC proliferation, as measured by BrDU incorporation. Mechanistically, the authors provide in vitro evidence that loss of EVL causes a decrease in VEGFR2 internalization and ERK phosphorylation. The data are clearly presented and the manuscript is easy to read and follow. Below are my comments how to improve the manuscript.

Reply: We would like to thank the reviewer for his/her constructive criticism of our manuscript and hope that the additional data together with the changes to the text and the comments listed below satisfactorily address the concerns raised.

1. The authors need to tone down their statement (e.g. the first sentence of the discussion) that EVL is an important regulator of developmental angiogenesis in the postnatal retina. Clearly, only a mild delay of angiogenic outgrowth is visible during a narrow time window. At P7 the retinal outgrowth is similar to wildtype embryos.

Reply: We agree with the suggestion of the reviewer and have adjusted our statements in the results and discussion sections accordingly (page 8, 9).

2. Does the proliferation defect persist after P5 or is BrDU incorporation normal again at P7?

Reply: We have not observed obvious differences between P7 retinas from wild-type and EVL-deficient animals and the radial vascular outgrowth relative to the retinal radius was similar between the two groups at this point in time (Figure 3 A+B). Our cellular data suggest that EVL deficiency impairs VEGF-receptor 2 internalization and downstream signaling. Given that the magnitude of hypoxia/VEGF-release decreases in the postnatal mouse retina from P0 to P7, it seems logical that the impact of EVL-deficiency on endothelial proliferation and sprouting may be more prominent at earlier points in time, e.g. P3/P5 vs. P7. We have added this information in the discussion (page 11) and in the results section on page 6.

3. It is not clear whether the delay in outgrowth is due to a general delay in postnatal development. Is the body weight of mutant and wildtype embryos comparable?

Reply: We agree that this is an important point. In the revised version of the manuscript, we have now included a supplemental figure (EV5A) with the body weights of the analyzed wild-type and EVL-deficient pups at P3 and P5. At the given points in time, the body weights of the two groups were virtually identical (P3: WT (2.10g) vs. EVL^{-/-} (2.12g); P5 WT (3.00g) vs. 2.92g)), indicating that a global delay in postnatal development is most likely not the underlying reason for the delayed postnatal vascular outgrowth in EVL-deficient animals. We have included this information in the materials and methods section on page 13. We have now also included the body weights of the analyzed endothelial cell specific EVL knockout mice (EVL^{ΔEC}) and littermate controls (EVL^{fl/fl}) at P5 (Figure EV5B). Again, there was no significant difference in body weight between the two groups (EVL^{fl/fl} (2.81g) vs. EVL^{ΔEC} (2.76g)).

4. The data seem rather "spread out". For example, figure 1 has little content, panel A could be moved to supplementary figures and expression of EVL could be moved from supplemental figure to main figure. Same is true for figure 4.

Reply: As suggested, we have now moved Figure 1A to the supplementary figures and moved the immunofluorescence images from the supplemental figure to the main figure of the revised manuscript.

5. The data on EVL expression in supplemental figure 1 only show high magnification views. How about overview images? Is EVL enriched in tip cells?

Reply: Initially, we aimed to investigate the expression of EVL in overview images and also in tip cells vs. stalk cells or more mature vessels in the center of the retina. Unfortunately, this was technically very difficult, because EVL is highly expressed in microglia and several classes of neurons including those of the innermost retinal layer just underneath the superficial vascular plexus and the tip cells, which have a very low dimension along the Z-axis. Given that retinal whole mounts are not perfectly flat per se and tend to bend during preparation, we were only able to obtain meaningful images of EVL expression in endothelial cells/vessels when using high magnification objectives with a small depth of field. We had mentioned this aspect in the discussion section on page 9. Given the difficulties with the multiple EVL-expressing cell types in retinal whole mounts, we therefore investigate the subcellular distribution of EVL in migrating endothelial cells *in vitro*, where EVL was concentrated at the tips of filopodia-like structures. Consistent with the suggested role of EVL in retinal tip cells, filopodia numbers at the vascular front of the postnatal mouse retina were significantly reduced in EVL-deficient animals.

6. The transcriptome analysis of EVL loss of function (Figure 4) needs to be substantiated. The authors only present a pathway analysis, but do not provide information on individual misregulated genes. Which genes are most highly changed in their expression and do the authors observe differences in the expression of these genes using antibody stainings in the retina? Related to this: The authors find a reduction in tip cell numbers in mutant retinas. They need to substantiate this finding by performing a staining for tip cell markers, e.g. ESM1.

Reply: Supplemental Table 1 contains the full data set of all analyzed genes, including the fold-change of the misregulated genes. We had intended to submit the supplemental Table 1 as Excel file to enable sorting the individual genes according to their change (column O: “log₂_fold_change”) etc. Unfortunately, however, the submission system transformed the Excel file into a PDF file, without this option. As suggested, we have now substantiated the transcriptome analysis and extracted those genes, which were markedly down-regulated ($\log_2(\text{fold-change}) < -0.5$) and associated with the most significantly changed GO-terms “blood vessel development” and “blood vessel morphogenesis”. These downregulated genes are shown in the new supplemental table EV4 and included cyclooxygenase 2 (PTGS2), serpine 1, paxillin and the tip cell marker ESM1, which are all known to promote angiogenesis (Boscher et al., 2019; Gately and Li, 2004; Rocha et al., 2014; Takayama et al., 2016). Furthermore, we confirmed the downregulation of these genes in EVL-deficient mouse endothelial cells vs. wild-type cells (Figure D in transcriptome analysis figure) and downregulation of Esm1 and Ptgs2 was also confirmed in HUVEC transfected with EVL-specific or control siRNA (supplemental Figure EV5). This information is also given in the results section on page 7. We had also intended to quantify the tip cell numbers in mutant retinas by ESM1 staining as suggested. Unfortunately, we had to transfer our EVL-deficient mice to a new animal facility and the required embryo transfer failed twice. Therefore, we apologize that we were unable to perform this experiment before the deadline of the revision. However, it is standard in the field to identify endothelial tip cells by their characteristic morphology and the significant reduction of Esm1 RNA levels in EVL-deficient mouse endothelial cells (Figure 5D) and siRNA-treated human endothelial cells (Figure EV4A) supports our conclusion that EVL-deficiency reduces tip numbers in mutant retinas.

7. The results in cultured cells appear more pronounced compared to the in vivo data. Good pERK antibodies are available for staining retinal ECs. Is pERK changed in mutant retinas? Are there differences between P5 and P7? How do the authors explain apparently normal retinal

outgrowth after P7 even though the authors claim that VEGFR2 signaling and ERK phosphorylation are reduced?

Reply: In cultured endothelial cells, EVL-deficiency decreased the VEGF-induced VEGFR2 and ERK phosphorylation by ~20-25%, which at least in part explains the delayed angiogenic sprouting in the mouse retina. The in vitro results appear indeed somewhat more pronounced than some of the effects observed in vivo, e.g. radial outgrowth and filopodia numbers in EVL^{ΔEC} retinas at P5. Interestingly, the delay was most pronounced in P3 retina, smaller in the P5 retina and vanished in the mostly normoxic P7 retina, suggesting a role of EVL particularly in the early stages of retinal angiogenesis, where levels of hypoxia/VEGF-release are high. This may also explain the apparently normal retinal outgrowth after P7.

We agree with the suggestion of the reviewer to address VEGFR2 signaling in vivo. Indeed, we have previously attempted to investigate VEGFR2- and ERK-phosphorylation in the mouse retina. However, even in wild-type animals this turned out to be variable and technically challenging, making the interpretation of the results very difficult. This is likely due to the fact that both phosphorylation events are very transient in nature, while fixation of the enucleated and therefore non-perfused mouse eye is a slowly progressing process that may distort the endogenous signal. Moreover, ERK phosphorylation is not only caused by VEGFR2 signaling and other pathways but also by stress hormones and mechanical strain, which further complicated the analysis. Therefore, we focused on the analysis of VEGFR2 signaling in cell culture, which can be better controlled.

8. The EC specific knockout figure should be moved after figure 3, as the authors only analyze tip cell and filopodia numbers in the EC specific EVL knockout mice. As the order is right now, I asked myself why the authors do not analyze filopodia in the EVL knockout mice, since they introduce EVL as an important player in filopodial dynamics (I had to wait to the last figure).

Reply: As requested, we have moved the EC-specific knockout figure, which improved the flow of the paper.

9. The abstract is confusing for non-experts. The authors introduce Ena/VASP proteins followed by stating that EVL and VASP, but not Mena are expressed in ECs. For the non-expert it is not clear how EVL and Mena are related to Ena/VASP from the current wording of the abstract.

Reply: We have revised the abstract to clarify this point.

10. The authors should mention in the discussion that a study in zebrafish found that EC filopodia are dispensable for EC guidance and that their ablation only weakly affected EC migration (Phng et al., 2013, Development).

Reply: We have now included this finding of the mentioned Development paper on page 9 of the discussion section.

References

- Benz, P.M., C. Blume, S. Seifert, S. Wilhelm, J. Waschke, K. Schuh, F. Gertler, T. Munzel, and T. Renne. 2009. Differential VASP phosphorylation controls remodeling of the actin cytoskeleton. *J Cell Sci.* 122:3954-3965.
- Boscher, C., V. Gaonac'h-Lovejoy, C. Delisle, and J.P. Gratton. 2019. Polarization and sprouting of endothelial cells by angiopoietin-1 require PAK2 and paxillin-dependent Cdc42 activation. *Mol Biol Cell.* 30:2227-2239.
- Gately, S., and W.W. Li. 2004. Multiple roles of COX-2 in tumor angiogenesis: a target for antiangiogenic therapy. *Semin Oncol.* 31:2-11.
- Ha, C.H., and Z.G. Jin. 2009. Protein kinase D1, a new molecular player in VEGF signaling and angiogenesis. *Mol Cells.* 28:1-5.
- Janssens, K., L. De Kimpe, M. Balsamo, S. Vandoninck, J.R. Vandenneede, F. Gertler, and J. Van Lint. 2009. Characterization of EVL-I as a protein kinase D substrate. *Cell Signal.* 21:282-292.
- Lambrechts, A., A.V. Kwiatkowski, L.M. Lanier, J.E. Bear, J. Vandekerckhove, C. Ampe, and F.B. Gertler. 2000. cAMP-dependent protein kinase phosphorylation of EVL, a Mena/VASP relative, regulates its interaction with actin and SH3 domains. *J Biol Chem.* 275:36143-36151.
- Rocha, S.F., M. Schiller, D. Jing, H. Li, S. Butz, D. Vestweber, D. Biljes, H.C. Drexler, M. Nieminen-Kelha, P. Vajkoczy, S. Adams, R. Benedito, and R.H. Adams. 2014. Esm1 modulates endothelial tip cell behavior and vascular permeability by enhancing VEGF bioavailability. *Circ Res.* 115:581-590.
- Takayama, Y., N. Hattori, H. Hamada, T. Masuda, K. Omori, S. Akita, H. Iwamoto, K. Fujitaka, and N. Kohno. 2016. Inhibition of PAI-1 Limits Tumor Angiogenesis Regardless of Angiogenic Stimuli in Malignant Pleural Mesothelioma. *Cancer Res.* 76:3285-3294.
- Wong, C., and Z.G. Jin. 2005. Protein kinase C-dependent protein kinase D activation modulates ERK signal pathway and endothelial cell proliferation by vascular endothelial growth factor. *J Biol Chem.* 280:33262-33269.

Dear Dr. Benz,

Thank you for submitting your revised manuscript to EMBO Reports. We have now received comments of both of the original referees, which are included below. In addition, I sought arbitrating advice from a good expert in the field whose opinion we trust.

I apologize for this unusual delay in getting back to you. It took longer than anticipated to receive the referee reports due to disturbances caused by the ongoing SARS-COV-2 pandemic.

Referee #1, who is an expert in endocytosis/signaling, finds that his/her concerns were satisfactorily addressed. However, referee #3, who is an angiogenesis expert, finds that the revision did not satisfactorily address his/her concerns regarding the in vivo aspect of the study, and does not find the revised manuscript suitable for publication in this journal. This was also reflected in his/her rankings of the reviewer evaluation table. Given these comments and as mentioned above, I consulted with a recognized expert in the field, to get further input specifically on these concerns. I am afraid the arbitrating advisor, whose comments are also copied below, agrees with the concerns of referee #3 and finds that the in vivo data does not sufficiently support for the proposed role of EVL in VEGFR2 internalization/angiogenesis in the retina and thus, does not recommend publication here. Given such comments from these experts who are also experienced reviewers, we cannot offer to publish your manuscript.

You will note that arbitrating advisor also provided a complete report on the study. I would like to emphasize that the advisor's additional concerns, which were not raised by the referees in the first review round, did not form a basis for the decision.

That being said, we recognize that your findings will be of value to the field. Therefore I discussed your manuscript with Andrea Leibfried, the executive editor of Life Science Alliance (a.leibfried@life-science-alliance.org). I am happy to say that Andrea can offer publication of this work in Life Science Alliance. Andrea would expect further changes to the manuscript text and data representation in response to the remaining concerns. The newly raised concerns of the additional reviewer do not need to get addressed experimentally, but a response should get provided with adequate changes to the manuscript. Editorial process and publication is quite swift at Life Science Alliance. Andrea would be happy to answer any questions.

I realize that this negative decision will be disappointing, especially after this long peer review process, but I am afraid that the reports at hand did not allow for a different conclusion in this case. I hope you will view the possibility of a transfer favourably. If this is the case, please use the link below to transfer the manuscript directly. I again apologize for this unusual delay.

Kind regards,

Deniz Senyilmaz Tiebe

Deniz Senyilmaz Tiebe, PhD
Editor
EMBO Reports

Referee #1:

The authors have addressed all my concerns appropriately and I am happy to support publication of this paper in EMBO Reports.

Referee #3:

Unfortunately, the authors did not perform several of the experiments I suggested to substantiate their findings in the retina model. They did not perform pERK staining (it is well established in the mouse retina, e.g. Pontes-Quero, et al., Rui Benedito, High mitogenic stimulation arrests angiogenesis, Nature Communications 2019) or analyze BrDu incorporation at P7. They did not perform ESM1 staining or another tip cell marker in the retina (because of embryo transfer problems). The reason for the observed differences in the phenotype between P5 and P7 are not clear. Therefore, the retina data remain weak.

Referee #4: (the arbitrating advisor)

In this paper the authors demonstrate that EVL, a regulator of actin dynamics, regulates VEGFR2 phosphorylation and endocytosis resulting in impaired angiogenesis evaluated in vitro and in vivo by analyzing the retinal vascularization. The data are clear but some additional controls are required. In particular the scenario envisaged by the authors (EVL deletion inhibits angiogenesis) could be explained by other mechanisms suggested by the same data here reported. Therefore the authors have to rule out some other possibilities.

Figure 1. Lower magnifications are required to have a general view of the expression. Furthermore it should be important to show EVL expression at the vascular front and plexus area of maturing retinal vascular tree

Fig 2. This figure shows the localization of EVL in different EC types. However, this sentence in the text body is not supported by the experiments: "EVL was concentrated at actin-rich stress fibers, focal adhesions, filopodia and the leading edge of lamellipodia in migrating MLEC". At least the co-staining with paxillin or vinculin is required to identify the focal adhesions. If this experiment confirms the authors' statement it should be shown the focal adhesions in EVL KO EC. Actually there is a strong connection between VEGFR2 and FAK that acts as a signal transduction molecule at sites of focal adhesion, which are instrumental in cell motility and in cell-substrate attachment. If the authors demonstrate that EVL co-stained with a FA adhesion marker, I'd ask to show the features of FA (number, size) in ECs plated on a provisional extracellular matrix (e.g. fibronectin or collagen I).

The authors show a reduced extension of filopodia in retina vasculature of EVL null mice. Is there any cross talk between EVL and Notch pathway, which exerts a relevant role of stalk/tip dynamics? For instance, does DLL4 expression is modified?

Figure 6A. To address the specificity of EVL on VEGF EC stimulation, the authors compare the proliferative effect of VEGF and fibronectin. I have some concerns about the use of fibronectin

and I suggest to compare the effect of VEGF with another growth factor. Actually, 1) ref 35 does not describe any proliferative effect of fibronectin; 2) EC are stimulated by VEGF as soluble molecule. Is it the same for fibronectin? or EC were plated on fibronectin. If it is the case the stimulatory conditions are different and could condition the results. Furthermore the left panel lacks the pictures of EC stimulated by fibronectin.

Fig 7. The authors show that EVL deletion inhibits VEGF-mediated VEGFR2 phosphorylation and its endocytosis. The data here reported are very important but required more controls and evidences. Besides the internalization assay proposed I suggest an alternative approach based on the biotin use, which allows a best analysis of the process dynamics (eg 10.1371/journal.pbio.1000025). This approach will give further insights on the effect of EVL on the amount of VEGFR2 present on plasmamembrane.

A second issue related to this figure is the need to know if the lack of EVL inhibits the phosphorylation of other Tyrosines.

Third, does EVL deletion reduce VEGFR2 in retina phosphorylation?

A general failure of the manuscript is mechanism by which EVL regulates VEGFR2 endocytosis. This point is well addressed in the discussion. Because EVL regulates actin function, a general statement on impact of actin in endocytosis process should better set the problem. Furthermore I invite the authors to comment the caveolae-mediated endocytosis, which is another mechanisms of VEGFR2 internalization, and its putative link with EVL.

Minor criticisms

Change ref 4, 12-14 with original papers (e.g. PMID:12879061, 16940438, 9529250, 12892710, 9630219

Additional comments: the difficulties to perform the suggested in vivo experiments (eg the in vivo VEGFR2 phosphorylation has been analyzed by other authors supporting its feasibility) together the lack of a mechanism connecting EVL with VEGFR2 (I underlined that the authors well addressed this point in the discussion and I suggest to put their observations in the more general context of actin role of endocytosis) dampens my enthusiasm on the paper.

** As a service to authors, EMBO Press provides authors with the ability to transfer a manuscript that one journal cannot offer to publish to another journal, without the author having to upload the manuscript data again. To transfer your manuscript to another EMBO Press journal using this service, please click on Link Not Available

Dr. Peter Benz · Goethe Universität, FB Medizin · Theodor-Stern-Kai 7 · 60590 Frankfurt · Germany

Dr. Deniz Senyilmaz Tiebe
Monitoring Editor
EMBO Reports

Re: EMBO Reports manuscript EMBOR-2019-48961-T, "EVL regulates VEGF receptor 2 internalization and signaling in developmental angiogenesis"

Dear Dr. Senyilmaz Tiebe,

Thank you very much for your editorial letters dated August 27 and October 10, 2019.

We are very grateful for the opportunity to submit a revised version of our manuscript and for the helpful suggestions from the reviewers. In the revised manuscript we included additional experimental data and addressed all of the points raised by the two referees. Our detailed point-by-point response to their comments is given below. The RNA-Seq data from our study have been uploaded to the Gene Expression Omnibus database, but the validation process is currently ongoing and we will provide the corresponding GSE identifier once it is available.

Our manuscript has clearly been improved by addressing the reviewers' critiques and contains significant novel findings that we hope will be interesting for the readers of *EMBO Reports*. We are looking forward to hearing from you.

On the behalf of all authors,

Sincerely yours

Dr. Peter Benz

**Zentrum für Molekulare
Medizin**

Institute for Vascular Signalling
Direktor: Prof. Dr. I. Fleming

Dr. Peter Benz
Gruppenleiter
Tel.: +49-69-6301-86879
Fax: +49-69-6301-86880
benz@vrc.uni-frankfurt.de

Datum: 27.02.2020

www.sfb834.de

Postanschrift
Inst. for Vascular Signalling
Haus 25B, 2. OG
Theodor-Stern-Kai 7
60590 Frankfurt

Sekretariat
Tel.: 069-6301-6052
Fax: 069-6301-7668

WWW
www.vrc.uni-frankfurt.de
www.ivs.uni-frankfurt.de

Dear Dr. Benz,

Thank you for contacting us regarding the recent decision taken on your manuscript.

I would like to reiterate that we, like referees, did appreciate the amount of work that went into revision and that the manuscript was improved. However, referee #3 found that relevance of the findings in the retina was elusive, which was a concern also shared by the arbitrating advisor. Although we were aware of the technical difficulties you noted in the response to the referees and the additional in vitro work, given the input we received, we had no other choice but to conclude that we could not go ahead with the manuscript for this journal in the absence of data substantiating the in vivo relevance of the findings.

I appreciate that you are willing to address the remaining points of reviewer #3 experimentally. We would be happy to reconsider the manuscript with such new data. However, please note that we require strong support from the referees to consider publication here. It is this aspect that is more difficult to assess at this stage.

Thank you again for the opportunity to consider your manuscript. I am looking forward to your revision.

Kind regards,

Deniz Senyilmaz Tiebe

--

Deniz Senyilmaz Tiebe, PhD

Editor

EMBO Reports

Point by Point Reply**Referee #1:**

The authors have addressed all my concerns appropriately and I am happy to support publication of this paper in EMBO Reports.

Reply: We would like to thank the reviewer for his/her constructive criticism and for supporting the publication of our study.

Referee #3:

Unfortunately, the authors did not perform several of the experiments I suggested to substantiate their findings in the retina model. They did not perform pERK staining (it is well established in the mouse retina, e.g. Pontes-Quero, et al., Rui Benedito, High mitogenic stimulation arrests angiogenesis, Nature Communications 2019) or analyze BrDu incorporation at P7. They did not perform ESM1 staining or another tip cell marker in the retina (because of embryo transfer problems). The reason for the observed differences in the phenotype between P5 and P7 are not clear. Therefore, the retina data remain weak.

Reply: We understand the disappointment of the reviewer for not having conducted several of his/her suggested experiments. We agree that the suggested experiments are well suited to substantiate our findings and after the embryo transfer had finally worked, we have conducted all remaining experiments. We hope that the additional data together with the changes to the text and the comments listed below satisfactorily address the concerns raised.

Does the proliferation defect persist after P5 or is BrDU incorporation normal again at P7?

Reply: As suggested, we have now analyzed endothelial cell proliferation in P7 retinas from wild-type and EVL-deficient animals and there was no significant difference between the two groups. The normal endothelial cell proliferation in EVL-deficient animals at P7 is in agreement with the normal radial vascular outgrowth at this point in time (Figure 3 A+B) and supports the concept that EVL deficiency impairs VEGF-receptor 2 internalization and downstream signaling. Given that the magnitude of hypoxia/VEGF-release decreases in the postnatal mouse retina from P0 to P7, it seems logical that the impact of EVL-deficiency on endothelial proliferation and sprouting may be more prominent at earlier points in time, e.g. P3/P5 vs. P7. We have added this information in Figure EV5C and in the discussion section on page 11.

The authors find a reduction in tip cell numbers in mutant retinas. They need to substantiate this finding by performing a staining for tip cell markers, e.g. ESM1.

Reply: Our transcriptome analysis of sorted endothelial cells from EVL-deficient P5 retinas revealed a markedly downregulated expression of the tip cell marker ESM1. We confirmed the downregulation of this gene in EVL-deficient mouse endothelial cells vs. wild-type cells (Figure 5D) and downregulation of ESM1 mRNA was also confirmed in HUVEC transfected with EVL-specific or control siRNA (supplemental Figure EV4A). As suggested, we have now

substantiated our data by including ESM1 stainings in P5 retinas of wild-type and EVL-deficient animals. Consistent with our transcriptome analysis and the reduced number of tip cells at the vascular front of postnatal EVL^{-/-} retinas, we detected a significant reduction of ESM1 protein levels in EVL-deficient animals at P5. We have added this important information in Figure 5E and 5F, and in the results sections on page 7.

7. The results in cultured cells appear more pronounced compared to the in vivo data. Good pERK antibodies are available for staining retinal ECs. Is pERK changed in mutant retinas?

Reply: Using the suggested antibodies from the Pontes-Quero/Benedito et al. Nature Communications 2019 publication, we performed the phospho-ERK immunostainings. Very similar to our in vitro assays, we observed a highly significant reduction of ERK1/2 phosphorylation in EVL-deficient retinal endothelial cells at the vascular front. We are grateful for the valuable advice of the reviewer and have added the citation to our manuscript. The new data is given in the revised Figure 8C and 8D and in the results section on page 9.

Referee #4: (the arbitrating advisor)

In this paper the authors demonstrate that EVL, a regulator of actin dynamics, regulates VEGFR2 phosphorylation and endocytosis resulting in impaired angiogenesis evaluated *in vitro* and *in vivo* by analyzing the retinal vascularization. The data are clear but some additional controls are required. In particular the scenario envisaged by the authors (EVL deletion inhibits angiogenesis) could be explained by other mechanisms suggested by the same data here reported. Therefore the authors have to rule out some other possibilities.

Reply: We would like to thank the reviewer for his/her constructive criticism of our manuscript and hope that the additional data together with the changes to the text and the comments listed below satisfactorily address the concerns raised.

Figure 1. Lower magnifications are required to have a general view of the expression. Furthermore it should be important to show EVL expression at the vascular front and plexus area of maturing retinal vascular tree.

Reply: Initially, we aimed to investigate the expression of EVL in overview images and also in tip cells vs. stalk cells or more mature vessels in the center of the retina. Unfortunately, this was technically challenging, because EVL is highly expressed in microglia and several classes of neurons including those of the innermost retinal layer just underneath the superficial vascular plexus and the tip cells, which have a very low dimension along the Z-axis. We were only able to obtain meaningful images of EVL expression in endothelial cells/vessels using high magnification objectives with a small depth of field. We had mentioned this aspect in the discussion section on page 9. Given the difficulties with the multiple EVL-expressing cell types in retinal whole mounts, we therefore investigated the subcellular distribution of EVL in migrating endothelial cells *in vitro*, where EVL was concentrated at the tips of filopodia-like structures. Consistent with the suggested role of EVL in retinal tip cells, filopodia numbers at the vascular front of the postnatal mouse retina were significantly reduced in EVL-deficient animals.

Fig 2. This figure shows the localization of EVL in different EC types. However, this sentence in the text body is not supported by the experiments: "EVL was concentrated at actin-rich stress fibers, focal adhesions, filopodia and the leading edge of lamellipodia in migrating MLEC". At least the co-staining with paxillin or vinculin is required to identify the focal adhesions. If this experiment confirms the authors' statement it should be shown the focal adhesions in EVL KO EC. Actually there are strong connections between VEGFR2 and FAK that acts as a signal transduction molecule at sites of focal adhesion, which are instrumental in cell motility and in cell-substrate attachment. If the authors demonstrate that EVL co-stained with a FA adhesion marker, I'd ask to show the features of FA (number, size) in ECs plated on a provisional extracellular matrix (e.g. fibronectin or collagen I).

Reply: As suggested, we have substantiated our findings by immunofluorescence microscopy studies with the focal adhesion marker paxillin. The new data is shown in Figure EV3A and supports the localization of EVL to focal adhesions. There is indeed a strong connection between VEGFR2, FAK and angiogenesis. On the one hand, FAK that is required

for VEGFA-induced angiogenesis and vascular permeability (Chen et al., 2012; Tavora et al., 2010) on the other hand FAK and its kinase activity regulate VEGFR2 transcription in angiogenesis (Sun et al., 2018). Given the role of Ena/VASP proteins in focal adhesion assembly and the link between FAK and VEGFR2, it is indeed tempting to speculate that an impaired EVL-FAK-VEGFR2 signaling axis may contribute to the defective angiogenesis in EVL-deficient retinas. Consequently, we have analyzed focal adhesions and FAK distribution in wild-type and EVL-deficient mouse lung endothelial cells, but we didn't observe obvious differences between the two groups. This is likely due to the fact that VASP is also expressed in these cells and that Ena/VASP proteins can, at least in part, compensate for each other (e.g. (Damiano-Guercio et al., 2020)). To address this point in more detail, more sophisticated studies in VASP/EVL-double deficient endothelial cells would be required, potentially with individual rescue strategies. Although this is certainly worthwhile investigating in more detail, those studies are clearly above the scope of the current manuscript. Furthermore, FAK has been shown to regulate *expression* of VEGFR2. However, VEGFR2 levels were similar in wild-type and EVL^{-/-} MLEC, indicating that the defect cannot be explained by alterations of receptor expression per se (**Fig. 7A**).

The authors show a reduced extension of filipodia in retina vasculature of EVL null mice. Is there any cross talk between EVL and Notch pathway, which exerts a relevant role of stalk/tip dynamics? For instance, does DLL4 expression is modified ?

Reply: We fully agree with the comment of the reviewer. Indeed, we initially focused on a potential cross talk between EVL and Notch pathway. However, our transcriptome analysis of retinal endothelial cells (Table EV1) didn't reveal obvious differences in DLL4 or Notch expression or expression of Notch downstream targets (Hey-1/2, Hes-1) in EVL-deficient cells. Thus, DLL4/Notch signaling seems to be independent of EVL and our data rather support the concept that EVL regulates developmental angiogenesis by modulating VEGF receptor 2 internalization and signaling. We have added this information in the results section on page 7-8.

Figure 6A. To address the specificity of EVL on VEGF EC stimulation, the authors compare the proliferative of effect of VEGF and fibronectin. I have some concerns about the use of fibronectin and I suggest to compare the effect of VEGF with another growth factor. Actually, 1) ref 35 does not describe any proliferative effect of fibronectin; 2) EC are stimulated by VEGF as soluble molecule. Is it the same for fibronectin? or EC were plated on fibronectin. If it is the case the stimulatory conditions are different and could condition the results. Furthermore the left panel lacks the pictures of EC stimulated by fibronectin.

Reply: As suggested, we have analyzed additional soluble growth factors and selected EGF for further studies, because EGF induced an increase in proliferation of wild-type endothelial cells (albeit considerably less pronounced than VEGF). Similar to VEGF treatment, however, EGF-stimulation failed to increase the proliferation of EVL-deficient cells (see below). This is not surprising given the evolving understanding that VEGFR2 membrane trafficking conforms, at least in some respect, that of EGFR (Horowitz and Seerapu, 2012) (Zhang and Simons, 2014). Indeed, both depend, at least in part, on actin dynamics and silencing of Mena impaired the clathrin-mediated endocytosis of the EGF receptor in HeLa cells (Vehlow et al., 2013). We have further detailed this aspect in the discussion section on page 11. We

apologize for the wrong citation of the pro-mitotic effect of fibronectin (ref 35) and have changed this in the revised version of the manuscript (page 8). Stimulation of endothelial cell proliferation by fibronectin depends on integrin signaling. However, similar to growth factor activation, fibronectin-mediated integrin engagement activates multiple growth-associated kinases including ERK1/2 (Wilson et al., 2003). We've included this control to show that the mitotic capacity of EVL-deficient endothelial cells is not generally impaired. This information is now provided on page 8. Given that fibronectin-mediated proliferation involves ERK phosphorylation, which is also a key driver of VEGF-induced endothelial proliferation, this suggests that ERK1/2 signaling is likely not impaired in EVL-deficient endothelial cells per se.

Fig 7. The authors show that EVL deletion inhibits VEGF-mediated VEGFR2 phosphorylation and its endocytosis. The data here reported are very important but required more controls and evidences. Besides the internalization assay proposed I suggest an alternative approach based on the biotin use, which allows a best analysis of the process dynamics (eg 10.1371/journal.pbio.1000025). This approach will give further insights on the effect of EVL on the amount of VEGFR2 present on plasmamembrane. A second issue related to this figure is the need to know if the lack of EVL inhibits the phosphorylation of other Tyrosines. Third, does EVL deletion reduce VEGFR2 in retina phosphorylation?

Reply: Cell surface biotinylation is certainly an established method to study receptor trafficking. However, in contrast to antibody feeding assays, it does not permit visualization of receptor location and clustering, which is especially useful to study VEGFR2 membrane dynamics (Sawamiphak et al., 2010). As suggested, we aimed to study VEGFR2 phosphorylation in the retina either with phosphospecific VEGFR2 antibodies or proximity ligation assays with a combination of receptor- and phospho-tyrosine-specific antibodies. However, this turned out to be variable, making the interpretation of the results difficult. We have, however, successfully analyzed the downstream ERK-phosphorylation in P5 wild-type and mutant retinas. Very similar to our in vitro assays, we observed a highly significant reduction of ERK1/2 phosphorylation in EVL-deficient retinal endothelial cells at the vascular front. The new data is given in the revised Figure 8C and 8D and in the results section on page 9.

A general failure of the manuscript is mechanism by which EVL regulates VEGFR2 endocytosis. This point is well addressed in the discussion. Because EVL regulates actin function, a general statement on impact of actin in endocytosis process should better set the problem.

Reply: To facilitate a better understanding for a broad readership we've added general statements about the importance of actin dynamics for the different forms of endocytosis and the involved steps. This information is given in the discussion section on page 11.

Furthermore I invite the authors to comment the caveolae-mediated endocytosis, which is another mechanisms of VEGFR2 internalization, and its putative link with EVL.

Reply: We are well-aware that actin dynamics are also implicated in caveolae-mediated endocytosis and that VEGFR2 can also be internalized via this route. In contrast, however, very little is known about a role of Ena/VASP proteins in caveolae or caveolae-mediated endocytosis. Since this is a tempting hypothesis, we analyzed the subcellular distribution of Ena/VASP proteins and caveolin-1 in endothelial cells. At least under the applied experimental conditions, however, we didn't observe a substantial co-localization of the proteins (see below), arguing against a major role of Ena/VASP proteins in caveolae. However, our very preliminary results certainly don't preclude future studies in this direction.

Minor criticisms

Change ref 4, 12-14 with original papers (e.g. PMID:12879061, 16940438, 9529250, 12892710, 9630219)

Reply: As suggested, we included the original publications and removed ref4. However, we thought to keep refs 12-14 because those reviews provide complementary information, e.g. about the role of Ena/VASP proteins in actin assembly (ref14; (Dent et al., 2011)), etc.

Additional comments: the difficulties to perform the suggested in vivo experiments (eg the in vivo VEGFR2 phosphorylation has been analyzed by other authors supporting its feasibility) together the lack of a mechanism connecting EVL with VEGFR2 (I underlined that the authors well addressed this point in the discussion and I suggest to put their observations in the more general context of actin role

of endocytosis) dampens my enthusiasm on the paper.

Reply: As mentioned above, we have now included in vivo phosphorylation analysis of ERK, downstream of VEGFR2 phosphorylation. Furthermore, we have now put our observations in the more general context of the role of actin dynamics in endocytosis (see above).

- Chen, X.L., J.O. Nam, C. Jean, C. Lawson, C.T. Walsh, E. Goka, S.T. Lim, A. Tomar, I. Tancioni, S. Uryu, J.L. Guan, L.M. Acevedo, S.M. Weis, D.A. Cheresh, and D.D. Schlaepfer. 2012. VEGF-induced vascular permeability is mediated by FAK. *Dev Cell*. 22:146-157.
- Damiano-Guercio, J., L. Kurzawa, J. Mueller, G. Dimchev, M. Schaks, M. Nemethova, T. Pokrant, S. Bruhmann, J. Linkner, L. Blanchoin, M. Sixt, K. Rottner, and J. Faix. 2020. Loss of Ena/VASP interferes with lamellipodium architecture, motility and integrin-dependent adhesion. *Elife*. 9.
- Dent, E.W., S.L. Gupton, and F.B. Gertler. 2011. The growth cone cytoskeleton in axon outgrowth and guidance. *Cold Spring Harb Perspect Biol*. 3.
- Horowitz, A., and H.R. Seerapu. 2012. Regulation of VEGF signaling by membrane traffic. *Cell Signal*. 24:1810-1820.
- Sawamiphak, S., S. Seidel, C.L. Essmann, G.A. Wilkinson, M.E. Pitulescu, T. Acker, and A. Acker-Palmer. 2010. Ephrin-B2 regulates VEGFR2 function in developmental and tumour angiogenesis. *Nature*. 465:487-491.
- Sun, S., H.J. Wu, and J.L. Guan. 2018. Nuclear FAK and its kinase activity regulate VEGFR2 transcription in angiogenesis of adult mice. *Sci Rep*. 8:2550.
- Tavora, B., S. Batista, L.E. Reynolds, S. Jadeja, S. Robinson, V. Kostourou, I. Hart, M. Fruttiger, M. Parsons, and K.M. Hodivala-Dilke. 2010. Endothelial FAK is required for tumour angiogenesis. *EMBO Mol Med*. 2:516-528.
- Vehlow, A., D. Soong, G. Vizcay-Barrena, C. Bodo, A.L. Law, U. Perera, and M. Krause. 2013. Endophilin, Lamellipodin, and Mena cooperate to regulate F-actin-dependent EGF-receptor endocytosis. *EMBO J*. 32:2722-2734.
- Wilson, S.H., A.V. Ljubimov, A.O. Morla, S. Caballero, L.C. Shaw, P.E. Spoerri, R.W. Tarnuzzer, and M.B. Grant. 2003. Fibronectin fragments promote human retinal endothelial cell adhesion and proliferation and ERK activation through alpha5beta1 integrin and PI 3-kinase. *Invest Ophthalmol Vis Sci*. 44:1704-1715.
- Zhang, X., and M. Simons. 2014. Receptor tyrosine kinases endocytosis in endothelium: biology and signaling. *Arterioscler Thromb Vasc Biol*. 34:1831-1837.

Dear Peter,

Thank you for submitting your revised manuscript. It has now been seen by one of the original referees.

As you can see, the referee finds that the study is significantly improved during revision and recommends publication. Before I can accept the manuscript, I need you to address some minor points below:

- Please address the remaining minor concerns of referee #3.
- Please provide the manuscript text in .docx format.
- Please provide the figures in individually and at production quality.
- We noticed that the EV tables are missing. Tables EV1-EV3 should be called Dataset EV1-EV3, please also correct the text callouts. The remaining table should be called Table EV1.
- Please provide 3-5 keywords for your study. These will be visible in the html version of the paper and on PubMed and will help increase the discoverability of your work.
- We noted that Hannes Schmidt is missing from the Author Contributions section. Please use initials only. There are 2 authors with initials HS, so one should be HSch and the other HSt.
- We notice the presence of 'data not shown' phrase on page 8, which is not allowed to be used as per journal policy. Please either show the data, or remove the statement.
- Please add a separate paragraph with the header "Conflict of Interest".
- All articles published beginning 1 July 2020, the EMBO Reports reference style changed to the Harvard style for all article types. Details and examples are provided at <https://www.embopress.org/page/journal/14693178/authorguide#referencesformat>. Please update the reference style accordingly.
- We notice that Fig 1 D+E callouts are missing from the text.
- Papers published in EMBO Reports include a 'synopsis' and 'bullet points' to further enhance discoverability. Both are displayed on the html version of the paper and are freely accessible to all readers. The synopsis includes a short standfirst summarizing the study in 1 or 2 sentences that summarize the paper and are provided by the authors and streamlined by the handling editor. I would therefore ask you to include your synopsis blurb and 3-5 bullet points listing the key experimental findings.
- In addition, please provide an image for the synopsis. This image should provide a rapid overview of the question addressed in the study but still needs to be kept fairly modest since the image size cannot exceed 550x400 pixels.
- Our production/data editors have asked you to clarify several points in the figure legends (see attached document). Please incorporate these changes in the attached word document and return it with track changes activated. I am aware that the comments were made on an earlier version of the manuscript, please use the attached document as a reference and perform the changes on the latest version of the text.

Thank you again for giving us to consider your manuscript for EMBO Reports, I look forward to your minor revision.

Kind regards,

Deniz

--

Deniz Senyilmaz Tiebe, PhD
Editor
EMBO Reports

Referee #3:

The authors addressed my concerns and provide high quality data on changes in pERK and ESM1 in EVL knockout retinas. I have just 2 small comments, which the authors might want to address:

1. The authors state in the discussion "Given that the magnitude of hypoxia/VEGF release decreases in the postnatal mouse retina from P0 to P7", but do not provide a reference for this statement. A reference is needed here.
2. The authors provide high quality data on changes in ESM1 and pERK upon loss of EVL. However, the quantification of these data eludes me. Quantifications in Figure 5F and 8D (and I realize now that a lot of other quantifications are provided in the same way) are (probably) given in percent and normalized to wildtype. The authors then show a cloud of data points around 100 for WT Embryos. How does this work, since everything should be normalized to WT in the first place? In addition, it would be better to report on the units of the actual values (e.g. mm² or mm) of the data, since this would provide more information than a percentage. For example, for the outgrowth of the vasculature (radius) in Figure 4B provide the data in mm instead of percent, for the area of pERK or ESM1 provide mm² instead of percent.

Point by Point Reply

Referee #3:

The authors addressed my concerns and provide high quality data on changes in pERK and ESM1 in EVL knockout retinas.

Reply: We would like to thank the reviewer for his/her constructive criticism and for the appreciation of our work.

I have just 2 small comments, which the authors might want to address:

1. The authors state in the discussion "Given that the magnitude of hypoxia/VEGF release decreases in the postnatal mouse retina from P0 to P7", but do not provide a reference for this statement. A reference is needed here.

Reply: We apologize for the oversight and have now added two references ([1, 2]) supporting our statement.

2. The authors provide high quality data on changes in ESM1 and pERK upon loss of EVL. However, the quantification of these data eludes me. Quantifications in Figure 5F and 8D (and I realize now that a lot of other quantifications are provided in the same way) are (probably) given in percent and normalized to wildtype. The authors then show a cloud of data points around 100 for WT Embryos. How does this work, since everything should be normalized to WT in the first place?

Reply: As detailed in the figure legends, the quantification of ESM1 and pERK levels is given in percent and normalized to wild-type littermates. In our original submission, we had performed the quantification as now suggested by referee #3 (all wild-type values being 100 or 1.0, respectively). However, in the previous revision, referee #1 insisted on adjusting the quantification/statistics as currently displayed.

Quote: "The normalisation to 1.0 was done in an inappropriate way thereby losing the error bars for the WT condition. This needs to be normalised by anchoring 1.0 to the mean of WT in PRISM. The problem with your approach of normalisation is that the stats program has to assume that all values for the WT are exactly the same (which they are not) and thus they will have an artificially low SD"

In addition, it would be better to report on the units of the actual values (e.g. mm² or mm) of the data, since this would provide more information than a percentage. For example, for the outgrowth of the vasculature (radius) in Figure 4B provide the data in mm instead of percent, for the area of pERK or ESM1 provide mm² instead of percent.

Reply: There are different ways of presenting radial outgrowth. We and others ([3-6]) prefer presenting relative outgrowth values rather than mm. This is in part due to the fact that absolute outgrowth values in mm strongly depend on the several parameters including body weight of individual animals and size of the analyzed eyes, as well as the mounting procedure of the flat mounts. This results in more variation of the absolute values as compared to the relative values, e.g. the ratio of vascular radius vs. retinal radius. Using percent also allows for an easier comparison of radial outgrowth between different genetically engineered animals and/or different postnatal points in time.

As detailed in the Materials and Methods section, the quantification of ESM1 and phospho-ERK within the retina was conducted with a IB4 related threshold-based mask and measured pixel intensities per area (mean grey values). Therefore, reporting mm² values would be not correct and we would need to report the pixel intensities per area instead. We believe that those cumbersome values would not provide any additional useful information, but would rather confuse the readers. Therefore, we would suggest to keep the analysis in its current form.

1. West H, Richardson WD, Fruttiger M (2005) Stabilization of the retinal vascular network by reciprocal feedback between blood vessels and astrocytes. *Development* **132**: 1855-62
2. Ruiz de Almodovar C, Lambrechts D, Mazzone M, Carmeliet P (2009) Role and therapeutic potential of VEGF in the nervous system. *Physiol Rev* **89**: 607-48
3. Weini C, Riehle H, Park D, Stritt C, Beck S, Huber G, Wolburg H, Olson EN, Seeliger MW, Adams RH, *et al.* (2013) Endothelial SRF/MRTF ablation causes vascular disease phenotypes in murine retinæ. *J Clin Invest* **123**: 2193-206
4. Weini C, Wasylyk C, Garcia Garrido M, Sothilingam V, Beck SC, Riehle H, Stritt C, Roux MJ, Seeliger MW, Wasylyk B, *et al.* (2014) Elk3 deficiency causes transient impairment in post-natal retinal vascular development and formation of tortuous arteries in adult murine retinæ. *PLoS One* **9**: e107048
5. Lee HW, Chong DC, Ola R, Dunworth WP, Meadows S, Ka J, Kaartinen VM, Qyang Y, Cleaver O, Bautch VL, *et al.* (2017) Alk2/ACVR1 and Alk3/BMPR1A Provide Essential Function for Bone Morphogenetic Protein-Induced Retinal Angiogenesis. *Arterioscler Thromb Vasc Biol* **37**: 657-663
6. Eilken HM, Dieguez-Hurtado R, Schmidt I, Nakayama M, Jeong HW, Arf H, Adams S, Ferrara N, Adams RH (2017) Pericytes regulate VEGF-induced endothelial sprouting through VEGFR1. *Nat Commun* **8**: 1574

Dr. Peter Benz
Goethe University
Theodor-Stern-Kai 7
Frankfurt am Main, Hessen 60596
Germany

Dear Peter,

Thank you for submitting your revised manuscript. I have now looked at everything and all is fine. Therefore I am very pleased to accept your manuscript for publication in EMBO Reports.

Congratulations on a very nice study!

When I was performing the final checks, I noticed that when I resize the synopsis image according to the online publication format, the labels on the image become very difficult to read (please see attached). Could you please increase the font and maybe make them bold? You can send me the file per email. Thanks.

Kind regards,

Deniz

Deniz Senyilmaz Tiebe, PhD
Editor
EMBO Reports

At the end of this email I include important information about how to proceed. Please ensure that you take the time to read the information and complete and return the necessary forms to allow us to publish your manuscript as quickly as possible.

As part of the EMBO publication's Transparent Editorial Process, EMBO reports publishes online a Review Process File to accompany accepted manuscripts. As you are aware, this File will be published in conjunction with your paper and will include the referee reports, your point-by-point response and all pertinent correspondence relating to the manuscript.

If you do NOT want this File to be published, please inform the editorial office within 2 days, if you have not done so already, otherwise the File will be published by default [contact: emboreports@embo.org]. If you do opt out, the Review Process File link will point to the following statement: "No Review Process File is available with this article, as the authors have chosen not to make the review process public in this case."

Should you be planning a Press Release on your article, please get in contact with emboreports@wiley.com as early as possible, in order to coordinate publication and release dates.

Thank you again for your contribution to EMBO reports and congratulations on a successful publication. Please consider us again in the future for your most exciting work.

THINGS TO DO NOW:

You will receive proofs by e-mail approximately 2-3 weeks after all relevant files have been sent to our Production Office; you should return your corrections within 2 days of receiving the proofs.

Please inform us if there is likely to be any difficulty in reaching you at the above address at that time. Failure to meet our deadlines may result in a delay of publication, or publication without your corrections.

All further communications concerning your paper should quote reference number EMBOR-2019-48961V5 and be addressed to emboreports@wiley.com.

Should you be planning a Press Release on your article, please get in contact with emboreports@wiley.com as early as possible, in order to coordinate publication and release dates.

Corresponding Author Name: Peter M. Benz

Manuscript Number: EMBOR-2019-48961V5